# Universality and kernel-adaptive training for classically trained, quantum-deployed generative models

## Abstract

The instantaneous quantum polynomial (IQP) quantum circuit Born machine (QCBM) has been proposed as a promising quantum generative model over bit-strings. Recent works have shown that the training of IQP-QCBM is classically tractable w.r.t. the so-called Gaussian kernel maximum mean discrepancy (MMD) loss function, while maintaining the potential of a quantum advantage for sampling itself. Nonetheless, the model has a number of aspects where improvements would be important for more general utility: (1) the basic model is known to be *not* universal - i.e. capable of representing arbitrary distributions, and it was not known whether it is possible to achieve universality by adding *hidden* (ancillary) qubits; (2) A fixed Gaussian kernel used in the MMD loss can cause training issues, e.g., vanishing gradients, as we demonstrate in this paper. For the former, we prove that for an $n$-qubit IQP generator, adding $n + 1$ *hidden* qubits makes the model universal. For the latter, we propose a *kernel-adaptive* training method, where the kernel is adversarially trained. We formally prove that such adaptive kernels have strictly greater discriminative power, and also show that in the kernel-adaptive method, the convergence of the MMD value implies convergence in distribution of the generator. We also analyze the limitations of the MMD-based training method. Finally, we verify the performance benefits of our contributions on a synthetic, parity-check dataset. The results show that the kernel-adaptive training method outperforms the Gaussian kernel w.r.t. the total variation distance between the generator and the data, and the advantage of the adaptive method becomes larger as the qubit number increases. These modifications and analyses shed light on the limits and potential of these new quantum generative methods, which could offer the first truly scalable insights in the comparative capacities of classical versus quantum models, even without access to scalable quantum computers.

## 1 Introduction

Quantum computing holds great promise for advancing machine learning, yet achieving a practical quantum advantage remains a significant challenge. While theoretical results have shown separations in learning capabilities, which are often grounded in cryptographic hardness assumptions (Liu et al., 2021; Gyurik & Dunjko, 2023), such proofs are not necessarily relevant in practical settings. In real-world applications, a model's value is ultimately measured by its empirical performance. However, current quantum hardware limitations make it practically impossible to evaluate a model, let alone train it, beyond a small qubit count. This constraint prevents any meaningful assessment of their performance on realistic-scale tasks.

Recently, a surprising result has proven that a specific quantum generative model can be efficiently trained and, to an extent, evaluated at scale. The model is Quantum Circuit Born Machine (QCBM) (Liu & Wang, 2018; Benedetti et al., 2019), with the Instantaneous Quantum Polynomial (IQP) circuit as the generator (Bremner et al., 2010; Nakata & Murao, 2014). The model prepares an $n$-qubit state using a parameterized IQP circuit, and measuring each qubit in the computational basis yields a bitstring of length $n$ sampled from a distribution that depends on the quantum state via the Born rule. We call this model IQP-QCBM. This model has been demonstrated to be efficiently trainable at scale

on classical hardware under a specific training objective—namely, the Maximum Mean Discrepancy (MMD) with a Gaussian kernel. This was demonstrated on high-dimensional datasets using models of up to $1\,000$ qubits Recio-Armengol et al. (2025); Recio-Armengol & Bowles (2025), achieving performance comparable to or better than that of classical counterparts. In terms of the capacity of quantum advantage, it is known that this model exhibits provable classical hardness when used as a sampler under standard complexity-theoretical assumptions (Bremner et al., 2010; 2016; Marshall et al., 2024).

Despite these promising features, the existing IQP-QCBM framework has two important limitations. First, the model as defined in Recio-Armengol et al. (2025) is known to be *non-universal* as a generative model over the Boolean hypercube: there exist simple distributions over $\{0, 1\}^n$ that cannot be represented by any $n$-qubit IQP-QCBM. Since universality is highly desirable to ensure broad applicability of the model, this raises the important question of whether universality can be restored by augmenting the model with hidden (ancillary) qubits, in analogy with hidden units in classical Boltzmann machines. Second, the training procedure is tied to a fixed Gaussian kernel in the MMD loss. We prove that this choice can lead to exponentially small MMD values and gradients for certain pairs of distributions, creating severe optimization challenges and limiting the discriminative power of the loss.

To address the above issues, we make the following contributions:

- **Achieving universality for IQP-QCBM.** We first augment the IQP-QCBM model with extra hidden qubits that are traced out before sampling, and then we prove that this IQP-QCBM model with hidden qubits is universal for distributions on $\{0, 1\}^n$: any target distribution can be represented *exactly* by an IQP-QCBM with $n$ visible and *at most* $n + 1$ hidden qubits. Since this is a worst-case scenario, we also show an example that, for each $n$, there exist distributions that can be represented with only one extra hidden qubit.

- **Kernel-adaptive training of IQP-QCBM.** We generalize the classical training framework for IQP-QCBMs beyond the fixed Gaussian kernel MMD. Taking the spectral representation of the MMD metric, we suggest that it suffices to parameterize the Fourier transformation (a spectral measure) of a stationary kernel over $\{0, 1\}^n$, which allows us to use non-Gaussian kernels in MMD. We parameterize the spectral measure with a neural network. Also, we propose to adaptively learn this neural network parameterization as a minimax optimization problem during training, which we call the *kernel-adaptive adversarial* training.

- **Limitation of MMD-based training.** We also analyze inherent limitations of MMD-based training, showing that there exist worst-case pairs of distributions with maximal total variation distance yet exponentially small MMD for *any* choice of kernel, leading to vanishing gradients regardless of the specific kernel-adaptive strategy.

- **Numerics.** Finally, we numerically validate the kernel-adaptive training on parity-check datasets designed to challenge the Gaussian kernel. Across increasing problem sizes, the adaptive scheme consistently achieves lower total variation distance than fixed-kernel baselines, with the performance gap growing with the number of qubits.

This paper is organized as follows. In Section 2, we discuss related work. In Section 3, we review the background on IQP-QCBMs. Section 4 presents our universality results for models with hidden qubits. In Section 5, we introduce kernel-adaptive training and provide theoretical justifications for it. Also, we provide limitations of MMD-based training in Section 6. The Section 7 contains our numerical experiments on benchmark datasets. We conclude with a discussion and directions for future work in Section 8.

## 2 RELATED WORK

**Quantum circuit Born machines with IQP circuits.** Quantum Circuit Born Machines (QCBMs) have been proposed as a flexible family of quantum generative models, in which classical data distributions are represented by measuring parametrized quantum states in the computational basis (Liu & Wang, 2018; Benedetti et al., 2019). A particularly attractive instance arises when the parametrized circuit is restricted to the special class, known as Instantaneous Quantum Polynomial-time (IQP) circuits (Shepherd & Bremner, 2008; Bremner et al., 2010; Nakata & Murao, 2014),

yielding the model studied in (Recio-Armengol et al., 2025). This model constitutes the first known example of a quantum generative model that is simultaneously trainable at scale with respect to a Gaussian maximum mean discrepancy (MMD) loss function, and supported by strong evidence of sampling hardness. Our work builds directly on this line of research, proving universality results for an extended IQP–QCBM architecture and generalizing its training framework beyond the fixed Gaussian-kernel MMD considered in prior work.

**Sampling hardness.** At the same time, we have strong arguments that sampling from generic IQP circuits is classically hard. In particular, there are families of IQP circuits for which even multiplicative-error weak simulation by a classical polynomial-time algorithm would collapse the polynomial hierarchy (PH) to the third level Bremner et al. (2010). Also, under standard average-case hardness assumptions, a classical algorithm that approximately samples the output distribution within constant total-variation distance would likewise collapse PH, originally to the third level Bremner et al. (2016), and more recently sharpened to the second level Marshall et al. (2024).

**Efficient training at scale.** Recio-Armengol et al. (2025) proposed to use the maximum mean discrepancy (MMD) metric with a Gaussian kernel as the training loss function for IQP-QCBM, and proved that, for the IQP circuit, the MMD loss can be efficiently estimated on a classical computer, which allows for training IQP-QCBM classically. Specifically, the spectral representation of MMD involves computing the expectation of Pauli-$Z$ observables (see Eq. (2)), which can be estimated efficiently on classical hardware. While the MMD loss enables large-scale training, using it with a fixed Gaussian kernel can limit its ability to sufficiently identify the discrepancy between distributions and may lead to vanishing MMD values and gradients, as we show in this paper. To address this, we generalize the MMD loss to use general, non-Gaussian kernels and prove that it can still be efficiently estimated on classical computers. Furthermore, we introduce a kernel-adaptive adversarial training scheme for IQP-QCBMs that adapts the kernel during the training. While trainable/implicit kernels have been explored in classical generative modeling Li et al. (2017; 2019); Arbel et al. (2018); Mroueh & Nguyen (2021), to our knowledge, we are the first to propose adaptive kernels for efficient training of the IQP-QCBM family.

**Universality problem.** Universality has been established for important families of classical generative models, including Restricted Boltzmann Machines (Roux & Bengio, 2008; Montúfar & Ay, 2011), GAN-style generators (Liu et al., 2017; Lu & Lu, 2020), diffusion/score-based models (Song et al., 2021; Lee et al., 2023), and quantum expectation value sampler for continuous distributions (Barthe et al., 2025; Shen et al., 2024). In contrast, the IQP-QCBM architecture as introduced in Recio-Armengol et al. (2025) is known to be non-universal over $\{0,1\}^n$, for instance, there exists a simple target distribution on $\{0,1\}^2$ that cannot be represented exactly by any 2-qubit IQP-QCBM. This non-universality limits the expressivity thereof. It was conjectured that adding hidden (ancillary) qubits could achieve universality in analogy with classical restricted Boltzmann machines. In this work, we resolve this open question by proving that an $n$-qubit IQP circuit augmented with extra hidden qubits is universal for distributions over $\{0,1\}^n$.

## 3 PRELIMINARIES OF IQP-QCBM

In this section, we review the definition of the IQP-QCBM model and recall two important results: its non-universality and its classical trainability with respect to the maximum mean discrepancy (MMD) metric.

### 3.1 IQP-QCBM MODEL

**Definition 1** (Quantum Circuit Born Machine (QCBM)). *A QCBM is a parameterized quantum generative model specified by: (1) A parameterized quantum circuit $U(\theta)$ acting on $n$ qubits, preparing the state $|\psi(\theta)\rangle = U(\theta)|0\rangle^{\otimes n}$; (2) Measurement in the computational basis yields bitstrings $x \in \{0,1\}^n$ with probability: $q_\theta(x) = |\langle x|\psi(\theta)\rangle|^2$. The model distribution $q_\theta$ is defined by sampling from this measurement distribution.*

In this paper, we consider the following parameterization of $U(\theta)$.

**Definition 2** (Parametrized instantaneous quantum polynomial (IQP) circuit Shepherd & Bremner (2008)). *A parametrized IQP circuit on $n$ qubits is a quantum circuit of the form $U(\theta) = H^{\otimes n} D(\theta) H^{\otimes n}$, where $H$ denotes the Hadamard gate, and $D(\theta)$ is the product of $L \in \mathcal{O}(\mathrm{poly}(n))$ physical diagonal gates: $D(\theta) = \prod_j e^{i\theta_j Z_{g_j}}, \theta_j \in [0, 2\pi)$, where $Z_{g_j}$ is a tensor product of Pauli-Z operators acting on a subset of qubits specified by the nonzero entries of $g_j \in \{0,1\}^n$.*

## 3.2 NON-UNIVERSALITY

**Definition 3** (Universality of generative models). *A family of generative models $Q$ is universal if for any target distribution $p(x), x \in \mathcal{X}$ ($\mathcal{X}$ is a topological space) and any precision $\varepsilon > 0$, there exist model $q_\theta \in Q$ and a parameter setting $\theta$ such that $d(p, q_\theta) \leq \varepsilon$, where $d(\cdot, \cdot)$ is a metric between probability distributions, e.g., total variation, Wasserstein.*

IQP-QCBMs are not universal, as shown in Recio-Armengol et al. (2025): $n$-qubit models cannot represent any distribution over the Boolean hypercube $\{0,1\}^n$. For example, the distribution $p = \left(\frac{1}{3}, \frac{1}{3}, \frac{1}{3}, 0\right)$ over $\{0,1\}^2$ cannot be represented by 2-qubit model. More generally, we notice that *no* distribution over $\{0,1\}^2$ with support size 3 is expressible by a 2-qubit IQP-QCBM, see derivations in Section A.3. This non-universality constrains the model's representational capacity, motivating the need to look at a universal extension of IQP-QCBM introduced in Section 4.

## 3.3 CLASSICAL TRAINING WITH MMD METRIC

We recap MMD's definition on IQP-QCBM and then show it can be estimated efficiently.

**Definition 4** (Maximum Mean Discrepancy (MMD)). *Given two distributions $p$ and $q$ over space $\mathcal{X}$, and a kernel $k : \mathcal{X} \times \mathcal{X} \to \mathbb{R}$ which induces a reproducing kernel Hilbert space (RKHS) $\mathcal{H}$, the MMD metric is:*

$$\mathrm{MMD}(p, q) = \sup_{f \in \mathcal{H}, \|f\|_{\mathcal{H}} \leq 1} \left(\mathbb{E}_{x \sim p}[f(x)] - \mathbb{E}_{y \sim q}[f(y)]\right), \tag{1}$$

*where RKHS norm $\|f\|_{\mathcal{H}}$ is defined with the kernel $k$ (Muandet et al., 2017).*

With the Gaussian kernel, i.e., $k(b, b') = \exp\left(-\sum_{i=1}^n |b_i - b'_i|/2\sigma^2\right), b, b' \in \{0,1\}^n$, the MMD metric of the IQP-QCBM model admits the following form (Rudolph et al., 2024):

$$\mathrm{MMD}^2(p, q_\theta) = \mathbb{E}_{\alpha \sim G_\sigma}\left[\left(\langle Z_\alpha \rangle_p - \langle Z_\alpha \rangle_{q_\theta}\right)^2\right], \tag{2}$$

where $Z_\alpha \in \mathbb{C}^{2^n \times 2^n}$ is a Pauli-$Z$ operator acting non-trivially on the qubits indexed by the nonzero entries of bitstring $\alpha \in \{0,1\}^n$, and $\langle Z_\alpha \rangle_{q_\theta} = \mathbb{E}_{b \sim q_\theta} \langle b|Z_\alpha|b\rangle$ ($q_\theta$ is the output distribution of IQP-QCBM). The distribution $G_\sigma$ is:

$$G_\sigma(\alpha) = (1 - p_\sigma)^{n-|\alpha|} p_\sigma^{|\alpha|}, \quad \alpha \in \{0,1\}^n, \tag{3}$$

with $p_\sigma = \frac{1}{2}\left(1 - \exp(-1/2\sigma)\right)$ and $|\alpha|$ is the Hamming weight of $\alpha$.

Crucially, the expectation $\langle Z_\alpha \rangle_{q_\theta}$ can be estimated efficiently with a classical algorithm (Nest, 2009; Recio-Armengol et al., 2025).

**Lemma 1.** *Given a parameterized IQP circuit $q_\theta$, an expectation value $\langle Z_\alpha \rangle_{q_\theta}$, and an error $\varepsilon \in \mathcal{O}(\mathrm{poly}(n^{-1}))$, there exists a poly(n) time classical algorithm that samples a random variable with standard deviation less than $\varepsilon$ that is an unbiased estimator of $\langle Z_\alpha \rangle_{q_\theta}$.*

Despite that, we can train the model classically w.r.t. Gaussian kernel MMD. However, this kernel choice might bring a limitation: the corresponding distribution $G_\sigma$ has an exponential tail w.r.t. the Hamming weight (see Eq. (3)), making it hard to distinguish two distributions when $\langle Z_\alpha \rangle_p - \langle Z_\alpha \rangle_{q_\theta}$ is concentrated on the large/small Hamming weights. As we will prove in Section 5, the distribution $G_\sigma$ is actually the *spectral measure* of the kernel $k$ (via Bochner's theorem), allowing for using non-Gaussian kernels in the training.

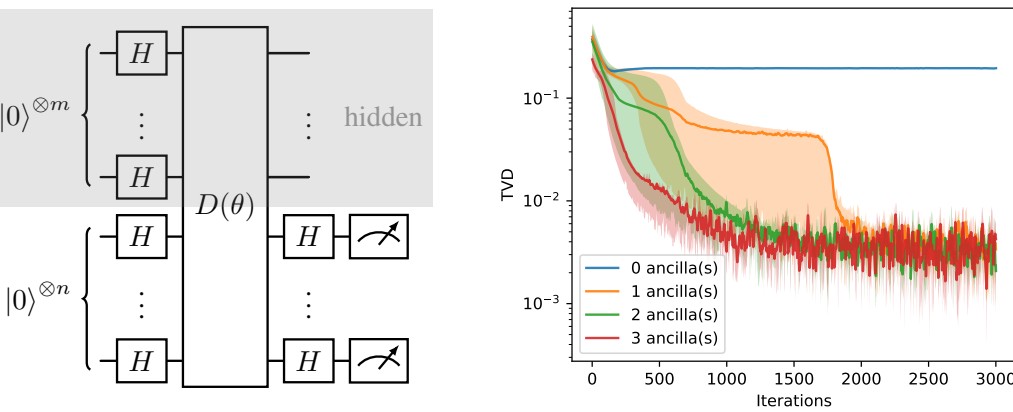

Figure 1: Left: schematic diagram of IQP-QCBM with hidden qubits. Right: example to show the expressivity enhancement via adding hidden/ancilla qubits for target distribution vector $p = \left(\frac{1}{3}, \frac{1}{3}, \frac{1}{3}, 0\right)$ over $\{0,1\}^2$. With just one hidden qubit, the total variation distance (TVD) between the generator and the target is greatly reduced over the training iterations.

## 4    UNIVERSALITY WITH HIDDEN QUBITS

In this section, we define the extended version of the original IQP-QCBM model by incorporating hidden qubits. We present two key results regarding their universality: one approximate-asymptotic and the stronger one exact. The last one, based on construction, is quite frugal, requiring only doubling the qubit number.

**Definition 5** (IQP-QCBM with hidden qubits). *A parameterized IQP circuit with hidden units is an IQP circuit in which a designated subset of qubits (called hidden units) is traced out prior to measurement. Let the full system have $m + n$ qubits, where the final $n$ qubits are designated as visible. The output distribution is obtained by taking the partial trace over the hidden qubits:*

$$q(x) = \mathrm{Tr}\left(\mathrm{Tr}_{hidden}\left(\rho\right)|x\rangle\langle x|\right), \tag{4}$$

*where $x \in \{0,1\}^n$ is a computational basis state, and $\rho$ is the quantum state produced by IQP.*

This construction draws inspiration from the structure of Boltzmann machines, where hidden neurons are marginalized out to increase expressivity. Similarly, introducing hidden qubits in IQP circuits allows for universal modeling of probability distributions. Our first proof is asymptotic.

**Lemma 2** (Asymptotic approximate universality with $\pi$-phases). *For any probability distribution $p$ over $\{0,1\}^n$, there exists a parameter setting $\theta = \{\theta_{j,k}\}_{j\in\{0,1\}^m, k\in\{0,1\}^n}$ of the IQP circuit acting on $m + n$ qubits such that the marginal distribution $q_\theta$ on the $n$ final qubits satisfies: $\delta(p, q_\theta) \in \mathcal{O}\left(\frac{1}{2^{m-n}}\right)$ where $\delta$ denotes the total variation distance between distributions.*

See the proof in Section A.1. The basic idea is to show that we can use ancillary registers to provide arbitrary counts of bitstrings in the output register, which, after normalization, achieves arbitrary frequencies of bitstrings, which then approximate the target distribution. This statement is not the most efficient possible, but utilizes only 0 and $\pi$ relative phases. By using all possible phases, a stronger result can be obtained.

**Theorem 1** (Exact universality). *For any target probability distribution $p$ over $\{0,1\}^n$, there exists an IQP circuit acting on $n$ visible qubits and $m = n + 1$ hidden qubits that produces $p$ exactly when we measure the final $n$ qubits.*

*Proof sketch.* First, we show that tracing out the hidden $m$ qubits gives a reduced density matrix:

$$\rho_2 := \mathrm{Tr}_{\mathrm{hidden}}(\rho) = \frac{1}{2^m}\sum_{k\in\{0,1\}^m}|\psi_k\rangle\langle\psi_k|, \tag{5}$$

where $|\psi_k\rangle = \frac{1}{\sqrt{2^n}}\sum_{y\in\{0,1\}^n} e^{i\theta_{k,y}}|y\rangle$. We show that it is possible to encode any 2-sparse distribution over $\{+,-\}$ basis in $|\psi_k\rangle$ by tuning the parameters. The probability of sampling string $b \in \{0,1\}^n$ is $|\langle\tilde{b}|\psi_k\rangle|^2$ with $|\tilde{b}\rangle = H^{\otimes n}|b\rangle$. Next, we prove that any probability distribution over

$\{0,1\}^n$ can be decomposed as a uniform mixture of at most $2^{n+1}$ 2-sparse distributions. Therefore, by choosing $m = n + 1$, we can represent any target distribution $p$ exactly using an IQP circuit with hidden qubits. See the Section A.2 for the complete proof. $\qquad\square$

Note that this universality result is a worst-case guarantee. However, for many distributions, adding only a single hidden qubit can increase the expressivity substantially. As demonstrated in Section 4, for the target distribution $p = \left(\frac{1}{3}, \frac{1}{3}, \frac{1}{3}, 0\right)$ over $\{0,1\}^2$, using just a single hidden qubit (the orange curve), instead of three additional ones, allows the IQP-QCBM to be trained to a satisfactory total variation distance from the target, compared with the baseline IQP model without hidden qubits. Moreover, we show that for some distributions not representable by the visible-only IQP-QCBM, adding just a single hidden qubit already yields an exact representation.

**Lemma 3** (Expressivity separation with one hidden qubit). *For every $n \in \mathbb{N}$ there exist a distribution $p$ on $\{0,1\}^n$, an $n$-qubit IQP-QCBM producing $q_\theta$, and an $(n+1)$-qubit IQP-QCBM with one hidden qubit producing $\tilde{q}_\omega$ such that:*

1. *There is a parameter value $\omega^*$ for the model with one hidden qubit satisfying $\tilde{q}_{\omega^*}(x) = p(x)$ for all $x \in \{0,1\}^n$.*

2. *There exists a constant $C > 0$ such that for every parameter setting $\theta$ of any visible-only $n$-qubit model, $\mathrm{TV}(p, q_\theta) \geq C$.*

The proof is given in Section A.4. In practice, however, we regard the number of hidden qubits as a model-capacity hyperparameter to be selected based on validation performance and specific data sets. Note that the hidden-qubit model can be efficiently trained on classical hardware in the same regime as the visible-only model. This follows from the fact that, similarly to Lemma 1, we can efficiently estimate expectation values of Pauli-$Z$ words with respect to the distribution produced by an IQP-QCBM with hidden qubits, as stated below.

**Remark 1.** *Given a parameterized IQP-QCBM model on $m$ hidden and $n$ visible qubits, an expectation value $\langle Z_\alpha \rangle_{q_\theta}$ and an error $\varepsilon \in \mathcal{O}(\mathrm{poly}(n^{-1}, m^{-1}))$, there exists a classical algorithm that requires $\mathrm{poly}(n, m)$ time to estimate $\langle Z_\alpha \rangle_{q_\theta}$ with at most $\varepsilon$ standard deviation of the estimation. Furthermore, this sample complexity can be refined to $\Omega(\mathrm{poly}(n + m)\frac{1}{\varepsilon^2} \log \frac{2}{\delta})$ for achieving any error smaller than $\varepsilon$ with at most $\delta$ probability, as proven in Section A.5.*

## 5 KERNEL-ADAPTIVE TRAINING

In this section, we introduce a kernel-adaptive adversarial training procedure for IQP-QCBM, designed to better align the model with the specific learning task of interest. First, we begin with an example to show that MMD with a fixed Gaussian kernel fails: there exist two distributions where the Gaussian MMD value decays exponentially while the total variation distance is constant. We note that the MMD distance is estimated via sampling, so exponential precision is not achievable in polynomial time. Second, we derive a generalized MMD expression applicable to arbitrary kernels, which can be efficiently estimated with classical algorithms. Third, we present the adversarial training method, which adaptively learns the kernel on the fly. Lastly, we establish convergence guarantees for our method.

### 5.1 LIMITATIONS OF THE GAUSSIAN MMD LOSS

We show that MMD with the Gaussian kernel might be ineffective in distinguishing probability distributions, resulting in a near-zero MMD value and vanishing gradients of the generator.

**Lemma 4** (Kernel choice matters). *For every $n \in \mathbb{N}$, there exist a target distribution $p$ on $\{0,1\}^n$, an IQP circuit on $n$ qubits with parameters $\theta$, and a setting $\theta'$ yielding distribution $q_{\theta'}$, such that: (1) $\mathrm{TV}(p, q_{\theta'}) \in \Omega(1)$; (2) For a Gaussian kernel $k_\sigma$: $\mathrm{MMD}^2_{k_\sigma}(p, q_{\theta'}), \left\| \nabla_\theta \mathrm{MMD}^2_{k_\sigma}(p, q_{\theta'}) \right\| \in \mathcal{O}(2^{-n})$; (3) For a characteristic kernel $\kappa$, there exists a constant $C > 0$ such that for all $n$: $\mathrm{MMD}^2_\kappa(p, q_{\theta'}), \left\| \nabla_\theta \mathrm{MMD}^2_\kappa(p, q_{\theta'}) \right\| \geq C$.*

See Section A.6 for the proof. The proof's key insight is: there exist distinct distributions $p$ and $q_{\theta'}$ such that the difference of their characteristic functions, $|\langle Z_\alpha \rangle_p - \langle Z_\alpha \rangle_{q_{\theta'}}|$, is supported on a

few bitstrings $\alpha$ of very small or very large Hamming weights. Now, consider the Gaussian kernel whose spectral measure $G_\sigma(\alpha)$ (Eq. (3)) decays exponentially with $n$ on both tails. In this case, $|\langle Z_\alpha \rangle_p - \langle Z_\alpha \rangle_{q_{\theta'}}|$ is supported on the tails of $G_\sigma(\alpha)$, which exponentially decays. In contrast, if we are allowed to deviate from the Gaussian, then we can choose a kernel $\kappa$ that concentrates on large or small Hamming weights, which prevents the exponential decay of the MMD value (w.r.t. $n$). The same argument applies to the gradient of MMD easily by the chain rule. Practically, one can choose the kernel $\kappa$ adaptively to allocate more spectral mass to the bitstring where the generator $\langle Z_\alpha \rangle_{q_{\theta'}}$ differs the most from the data $\langle Z_\alpha \rangle_p$.

## 5.2 MMD WITH ADAPTIVE KERNEL

We generalize MMD's expression beyond the Gaussian kernel based on the spectral representation of MMD, as a direct consequence of Bochner's theorem Bochner (1933).

**Lemma 5** (Spectral representation of MMD). *Let $X$ be a locally compact Abelian group and $p$, $q$ be probability distributions on $X$. Given a bounded, stationary kernel $k\colon X \times X \to \mathbb{R}$, and its Fourier transformation, $G$, which is a non-negative measure on the dual group $\widehat{X}$ (called the spectral measure of $k$). Then the squared Maximum Mean Discrepancy admits the spectral representation:*

$$\mathrm{MMD}_k^2(p, q) = \mathbb{E}_{\alpha \sim G}\left[|\phi_p(\alpha) - \phi_q(\alpha)|^2\right], \tag{6}$$

*where $\phi_p(\alpha) = \mathbb{E}_{x \sim p}[\exp(-i\alpha \cdot x)]$ and $\phi_q(\alpha) = \mathbb{E}_{x \sim q}[\exp(-i\alpha \cdot x)]$ are the characteristic functions of $p$ and $q$ respectively. See Muandet et al. (2017) for a proof.*

We notice that expectation values of Pauli-$Z$ both for target and IQP-QCBM produced distributions are characteristic functions of corresponding distributions: $\langle Z_\alpha \rangle_p = \phi_p(\alpha)$ and $\langle Z_\alpha \rangle_{q_\theta} = \phi_{q_\theta}(\alpha)$. This combined with Eq. (6), gives us:

**Theorem 2** (Generalized kernel MMD). *For any stationary and bounded kernel $k$ over $\{0, 1\}^n$, let $G$ be the Fourier transform of $k$. The MMD loss of IQP-QCBM can be expressed as:*

$$\mathrm{MMD}_G^2(p, q_\theta) = \mathbb{E}_{\alpha \sim G}\left(\langle Z_\alpha \rangle_p - \langle Z_\alpha \rangle_{q_\theta}\right)^2 \tag{7}$$

The above formulation generalizes the results in Recio-Armengol et al. (2025):

**Example 1.** *The Eq. (6) suggests that $G$ and $k$ are related via the Fourier transform. If we apply it to the Gaussian kernel, we obtain the following spectral mass:*

$$G(\alpha) = \frac{1}{2^n} \sum_{b \in \{0,1\}^n} (-1)^{b \cdot \alpha} \exp\left(-\|b\|^2/2\sigma^2\right) = (1 - p_\sigma)^{n - |\alpha|} p_\sigma^{|\alpha|} \tag{8}$$

*which exactly coincides with Eq. (3) with the same $p_\sigma$.*

**Remark 2** (Efficiency of generalized MMD estimation). *For the IQP-QCBM model, it is possible to construct an unbiased estimator of $\mathrm{MMD}^2$ and its gradients w.r.t. $\theta$ using an efficient classical algorithm. This is because (1) the expectation value $\langle Z_\alpha \rangle_{q_\theta}$ can be efficiently estimated by Lemma 1, and (2) $\mathrm{MMD}^2$ in Eq. (7) is a probabilistic mixture over these expectation values, which can be estimated efficiently by sampling bitstrings $\alpha \in \{0, 1\}^n$ from the measure $G$.*

## 5.3 ADVERSARIAL TRAINING WITH ADAPTIVE KERNEL

To leverage adaptive, task-dependent kernels in MMD based on generalized representations, we require kernels to be stationary and bounded. However, meaningful training necessitates an additional constraint—the kernel must be *characteristic*. A kernel $k$ over topological space $\mathcal{X}$ is characteristic if for any probability measures $\mu$ and $\nu$ in $\mathcal{X}$, the map $\mu \mapsto \int_{\mathcal{X}} k(x, \cdot) d\mu(x)$ is injective. This ensures distinct distributions yield unique kernel mean embeddings.

Being characteristic is essential for MMD loss, as it guarantees $\mathrm{MMD}^2(p, q) = 0$ *if and only if* $p = q$ (Muandet et al., 2017). For binary spaces $\{0, 1\}^n$, a kernel $k(t)$ is characteristic if and only if its spectral measure $G(\alpha)$ has full support ($\mathrm{supp}(G) = \{0, 1\}^n$), meaning $G(\alpha) > 0$ for all $\alpha \in \{0, 1\}^n$, see Fukumizu et al. (2008).

Instead of parameterizing the kernel, we model its spectral measure $G_\gamma$ using a critic network with

parameters $\gamma$. This leads to an adversarial training scheme for IQP-QCBMs, analogous to generative adversarial networks Goodfellow et al. (2020). We define the loss:

$$\mathcal{L}(p, q_\theta) := \max_\gamma \mathrm{MMD}^2_{G_\gamma}(p, q_\theta) \tag{9}$$

with the constrained optimization: $\min_\theta \mathcal{L}(\theta)$ s.t. $\sum_{\alpha \in \{0,1\}^n} G_\gamma(\alpha) = 1$ and $G_\gamma(\alpha) > 0 \quad \forall \alpha$.

In this min-max optimization setup, the critic $G_\gamma$ learns to identify the bitstrings at which $\langle Z_\alpha \rangle_p$ and $\langle Z_\alpha \rangle_{q_\theta}$ differ most by adjusting $G_\gamma(\alpha)$. Unlike the fixed-kernel training procedure proposed in Recio-Armengol et al. (2025), in our approach, we assign higher importance to the $\alpha$ values that are most relevant to the task of interest. This adaptive weighting intuitively suggests improved performance by focusing learning capacity on the most informative spectral features.

For adaptive kernel loss Eq. (9), it has been shown that the convergence in the loss value is equivalent to the convergence in distribution of probability measures (Simon-Gabriel et al., 2023) for classical learning algorithms in real spaces. Here, we specialize this argument for IQP-QCBM.

**Lemma 6** (Consistency with weak convergence). *Let $\{q_t\}_{t=1}^\infty$ be a sequence of probability distributions on $\{0,1\}^n$, and let $p$ be a fixed distribution on $\{0,1\}^n$. Let $G_\gamma : \{0,1\}^n \to (0,1)$ be any parametrized spectral measure with full support. Then the following equivalence holds:*

$$\lim_{t \to \infty} \mathcal{L}(q_t, p) = 0 \iff q_t \xrightarrow{d} p. \tag{10}$$

*where $\mathcal{L}$ is specified by Eq. (9).*

The lemma establishes a precise connection between the minimization of the trainable kernel MMD loss and the weak convergence of probability distributions: it tells us that if we imagine an idealized training process with an infinite number of steps, where at each step the model distribution $q_t$ is adjusted to reduce the MMD loss $\mathcal{L}(q_t, p)$ towards zero, then this iterative minimization guarantees that $q_t$ will eventually converge to $p$ in distribution. This means the MMD loss with adaptive kernel is not just a heuristic—it is *consistent* with weak convergence.

# 6 LIMITATIONS OF THE MMD LOSS

We now turn to the limitations of MMD-based losses for generative modeling. As shown in Lemma 4, the kernel choice is important: by emphasizing discrepancies at specific frequencies in the spectral representation of MMD, a suitable kernel can substantially affect training dynamics. This observation motivates adaptive kernel training, but even adaptivity is not a panacea. In fact, there exist distributions for which MMD training fails for *any* bounded characteristic kernel.

**Lemma 7** (Worst-case distributions). *There exist an absolute constant $b > 0$ and an $n_0 \in \mathbb{N}$ such that, for every $n \geq n_0$, there are distributions $p$ and $q$ on $\{0,1\}^n$ with $\mathrm{TVD}(p, q) = 1$ and, for any bounded characteristic kernel with spectral measure $G$ we have $\mathrm{MMD}_G(p, q) \leq 2^{-bn}$.*

*Proof sketch.* The proof is based on the existence (we prove this rigorously in Section A.8) of distributions with exponentially decaying characteristic functions for all $\alpha \neq \mathbf{0}$: $|\phi_p(\alpha)| \leq 2^{-c_1 n}$ and $|\phi_q(\alpha)| \leq 2^{-c_2 n}$ for fixed $c_1, c_2 > 0$, where $\mathbf{0}$ is the bitstring with zeroes in all positions and $\phi_p, \phi_q$ are the characteristic functions of the corresponding distributions. Then, for any bounded characteristic kernel with spectral measure $G$, the corresponding MMD can be bounded above as

$$\mathrm{MMD}^2_G(p, q) \leq \sup_{\alpha \neq \mathbf{0}} |\phi_p(\alpha) - \phi_q(\alpha)|^2 \leq 2^{-2n \min(c_1, c_2)}. \tag{11}$$

$\square$

This means that at any fixed frequency $\alpha$, the relevant signal $|\phi_p(\alpha) - \phi_q(\alpha)|$ is of order $2^{-\Omega(n)}$. Estimating such a quantity to constant relative accuracy typically requires $2^{\Omega(n)}$ samples, and even a perfect estimate yields an objective value that decays exponentially with $n$ (see Equation (11)), which creates severe optimization difficulties. In particular, for these worst-case distributions, under the mild assumptions the gradient with respect to the generator parameters also vanishes exponentially: $\|\nabla_\theta \mathrm{MMD}^2_G\| = 2^{-\Omega(n)}$, which implies that kernel-based training becomes ineffective for

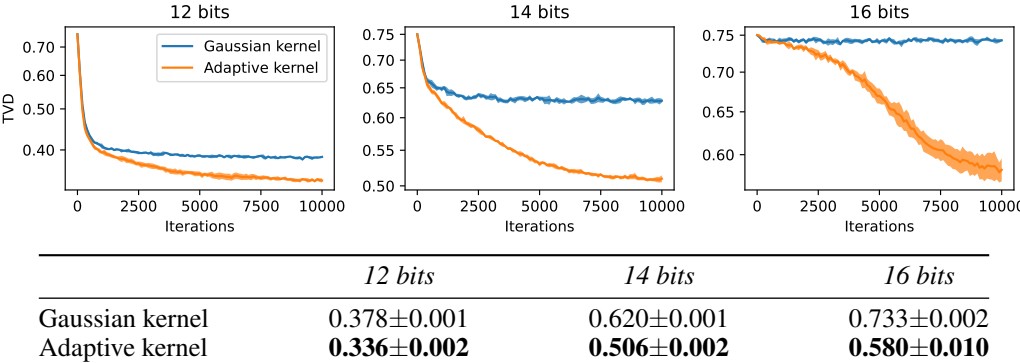

| | *12 bits* | *14 bits* | *16 bits* |
|---|---|---|---|
| Gaussian kernel | 0.378±0.001 | 0.620±0.001 | 0.733±0.002 |
| Adaptive kernel | **0.336±0.002** | **0.506±0.002** | **0.580±0.010** |

Figure 2: Total variation distance between generator distribution and ground truth distribution, when trained with different kernels (mean ± standard deviation over 5 runs) for 12-, 14-, and 16-bit synthetic parity-check datasets. Lowest achieved values are reported in the table. Bold indicates the best (minimal) TVD with statistical significance level of 0.01.

any kernel choice. This is a general limitation of the MMD metric, regardless of whether the generative model for training is classical or quantum. This issue is not unique to the IQP-QCBM model but rather a reflection of MMD's general insensitivity to certain distributions that can be artificially constructed. However, in practical scenarios, we expect that many real-world distributions can still be meaningfully distinguished using MMD with the adaptive training procedure.

## 7    NUMERICAL EXPERIMENTS

In this section, we present numerical experiments to evaluate the proposed kernel-adaptive training, employing synthetic datasets designed to illustrate scenarios where our method is beneficial.

**Dataset.**    We construct each synthetic dataset $\mathcal{D}$ by drawing samples from the null space of a parity-check matrix $H \in \{0, 1\}^{K \times n}$, i.e., for all $x \in \mathcal{D}$ and all $1 \leq k \leq K$, it holds that $\sum_{i=1}^{n} x_i h_i^{(k)} = 0$ (mod 2), where $h^{(k)}$ denotes the $k$-th row of $H$. The characteristic function of the underlying distribution $p$ of such a dataset differs from that of the uniform distribution $\mathcal{U}(\{0, 1\}^n)$ only at certain frequencies, namely the vectors $h^{(k)}$ and their XOR combinations. We construct such datasets over $\{0, 1\}^n$ with increasing $n = 12, 14, 16$, resulting in progressively harder tasks. Each dataset contains 2000 training samples and 50000 test samples. To challenge Gaussian kernels, we construct datasets whose distributions have only three non-trivial frequencies at high Hamming weight, $|\alpha| > \frac{n}{2}$. It can be shown that, for any fixed bandwidth $\sigma \geq 0$, the squared Gaussian kernel MMD decays exponentially with the model size $n$. Assuming that at initialization the generated distribution $q_\theta$ is close to $\mathcal{U}(\{0, 1\}^n)$, we expect both the magnitude of the Gaussian MMD loss and its gradient to vanish exponentially.

**Spectral Measure Parameterization.**    For our experiments, we employ a specific parametrization of the spectral measure $G_\gamma$ using a classic yet effective autoregressive model, the Fully Visible Sigmoid Belief Network (FVSBN) Neal (1992); Frey et al. (1997). More advanced architectures, such as NADE Larochelle & Murray (2011), MADE Germain et al. (2015), and Discrete Flows Tran et al. (2019), could also be considered as alternatives. The FVSBN models a factorized joint distribution over binary variables as follows:

$$G_\gamma(\alpha) = \prod_{i=1}^{n} \left[ p_\gamma(\alpha_i = 1 \mid \alpha_{<i}) \right]^{\alpha_i} \left[ 1 - p_\gamma(\alpha_i = 1 \mid \alpha_{<i}) \right]^{1-\alpha_i} \tag{12}$$

where the probability of each bit $\alpha_i$ is recursively computed from the previously sampled bits $\alpha_{<i} := (\alpha_1, \ldots, \alpha_{i-1})$. Specifically, the conditional probability is given by

$$p_\gamma(\alpha_i = 1 \mid \alpha_{<i}) = \varepsilon + (1 - 2\varepsilon)\,\mathrm{sigmoid}\Big(b_i + \sum_{r<i} W_{ir}(2\alpha_r - 1)\Big) \tag{13}$$

where the model is parameterized with parameters $\gamma = (W, b)$: $W \in \mathbb{R}^{n \times n}$ is a lower-triangular weight matrix, $b \in \mathbb{R}^n$ is a bias term. The small $\varepsilon = 10^{-6}$ value is to ensure numerical stability. FVSBN allows for both ancestral sampling of frequencies $\alpha$ and access to their log-probabilities $\log G_\gamma(\alpha)$, as required to compute the MMD loss gradients w.r.t. the parameters $\gamma$. Using the log-derivative trick, one can show that $\nabla_\gamma \widehat{\mathrm{MMD}}^2_{G_\gamma}(p, q_\theta)$ can be estimated as:

$$\frac{1}{K} \sum_{k=1}^{K} \nabla_\gamma \log G_\gamma(\alpha_k) \left( \langle Z_{\alpha_k} \rangle_p - \langle Z_{\alpha_k} \rangle_{q_\theta} \right)^2 \tag{14}$$

where $\alpha_k$ are frequencies sampled from $G_\gamma(\alpha)$.

**Results.** For each synthetic parity-check dataset, we use the same IQP generator, which includes all gates acting on up to six qubits. Its parameters are initialized to match the data covariance, as described in Recio-Armengol et al. (2025). We compare our adaptive training to a fixed Gaussian kernel with bandwidth $\sigma = 10^{-6}$ (approximately a uniform spectral measure). We tuned $\sigma$ as a hyperparameter and found that larger values degrade performance. For a fair comparison, $G_\gamma$ is initialized to the spectral measure of the Gaussian kernel with $\sigma = 10^{-6}$. For the adaptive training and the fixed Gaussian kernel, we use the same IQP-QCBM generator. The model's performance is assessed by the total variation distance (TVD) observed on the test sample, measured in every 100 training iterations.

The results are shown in Fig. 2. We clearly see that for all three problem sizes, the adaptive training outperforms the Gaussian training. Furthermore, the performance gap widens at larger bit counts. For 16 bits, while the Gaussian kernel training stagnates, the IQP generator improves and converges under adaptive training. As dimensionality increases, each minibatch covers an exponentially smaller fraction of the $2^n$ frequency space. Hence, a zero-initialised FVSBN draws the informative high-weight frequencies more rarely, which is visible in its slower convergence. To verify that our adaptive training generally outperforms Gaussian kernels regardless of the bandwidth, we repeated the training with different bandwidths $\sigma > 0$, or allowed the bandwidth to be dynamically updated during training. All these runs yield worse results than the $\sigma = 0$ cases, which is expected from the special construction of the datasets.

## 8 DISCUSSION AND CONCLUSION

In this work, we analyzed the prospects and limitations of the IQP-QCBMs as quantum generative models. In particular, we addressed two limitations. First, we established the universality of IQP-QCBM when hidden qubits are added, providing two proofs: a simple asymptotic version and an exact construction where adding $n + 1$ hidden qubits makes the $n$-qubit IQP generator universal. However, we emphasize that this construction serves primarily theoretical purposes and should not be interpreted as recommending large numbers of hidden units in practice. Our result instead illuminates hidden units and diagonal operations as architectural controls for balancing expressivity and trainability.

Second, we introduced an adversarial kernel-adaptive training method, which is classically efficient for IQP-QCBM. As we showed, this method addresses training issues in fixed-kernel MMD approaches. We provide theoretical guarantees that MMD convergence implies convergence in distribution under adaptive training. Experimental validation on the parity-check dataset demonstrates that this adaptive approach consistently reduces total variation distance compared to fixed Gaussian kernels, with performance gaps increasing as qubit count grows. While adaptive kernels can resolve training issues, we note that MMD-based training retains inherent limitations.

For future works, we plan to understand lower bounds on qubits required for hidden-model universality, implement the method for hidden-qubit architectures to quantify expressivity-trainability tradeoffs, and test kernel-adaptive training on more real-world datasets against fixed-kernel baselines.

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

# A APPENDIX

## A.1 PROOF OF LEMMA 2

*Proof.* The resulting quantum state after Hadamard gates to all $m + n$ qubits, a diagonal phase operator $D(\theta)$ with entries $e^{i\theta_{j,k}}$, and then another layer of Hadamards on the final $n$ qubits can be written as:

$$|\psi\rangle = I^{\otimes m} \otimes H^{\otimes n} \left( \sum_{j\in\{0,1\}^m} \sum_{k\in\{0,1\}^n} \frac{e^{i\theta_{j,k}}}{\sqrt{2^{m+n}}} |j\rangle |k\rangle \right) = \frac{1}{\sqrt{2^m}} \sum_{j\in\{0,1\}^m} |j\rangle |v(\theta_j)\rangle \quad (15)$$

where:

$$|v(\theta_j)\rangle := H^{\otimes n} \left( \frac{1}{\sqrt{2^n}} \sum_{k\in\{0,1\}^n} e^{i\theta_{j,k}} |k\rangle \right) \quad (16)$$

and $\theta_j := \{\theta_{j,k}\}_{k\in\{0,1\}^n}$ denotes the vector of phases for hidden unit $j$.

Note that each state $|v(\theta_j)\rangle$ can be made equal to any computational basis state $|s\rangle$ for $s \in \{0,1\}^n$ by choosing $\theta_{j,k} = \pi \times (k \cdot s)$. So, there exists a choice of $\theta_j \in \{0,\pi\}^n$ such that $|v(\theta_j)\rangle = |s\rangle$. Thus, for any output bitstring $s$, we can choose the set of $j \in \{0,1\}^m$ such that $v(\theta_j) = s$, effectively assigning a portion of the $2^m$ hidden units to output $s$. Then, the marginal distribution on the last $n$ qubits is:

$$q_\theta(s) = \frac{1}{2^m} |\{j \in \{0,1\}^m \mid v(\theta_j) = s\}|. \quad (17)$$

This shows that $q_\theta$ is a distribution over $\{0,1\}^n$ whose probabilities are rational numbers with denominator $2^m$, i.e., $m$-bit precision probabilities which can be chosen arbitrarily by tuning $\theta$. Thus, any target distribution $p$ over $\{0,1\}^n$ can be approximated well enough with increasing $m$. We prove the bounds for the approximation error in the total variation error below.

Let $p = (p_1, \ldots, p_{2^n})$ be the target distribution, then each component can be represented as:

$$p_j = \frac{i_j}{2^m} + \varepsilon_j \quad (18)$$

where $i_j \in \mathbb{N}_{\geq 0}$, and $\varepsilon_j \in \left[0, \frac{1}{2^m}\right]$ is the rounding error. Denote the total rounding reminder as $r := 2^m \sum_j \varepsilon_j \in \mathbb{N}_{\geq 0}$.

To construct a valid model probability distribution $q$ supported on $\{0,1\}^n$ with entries $q_j \in \left\{0, \frac{1}{2^m}, \frac{2}{2^m}, \ldots\right\}$ we redistribute this leftover $r$ over any $r$ components by incrementing their count by 1:

$$q_j = \begin{cases} \frac{i_j}{2^m} + \frac{1}{2^m} & \text{if } j \leq r \\ \frac{i_j}{2^m} & \text{if } j > r \end{cases} \quad (19)$$

Indeed, it is easy to verify that $q$ is a valid probability distribution since $\sum_j q_j = 1$. Then the total variation distance between the target distribution $p$ and the model distribution $q$ can be upper-bounded in the following way:

$$\delta(p,q) = \frac{1}{2} \sum_{j=1}^{2^n} |p_j - q_j| = \frac{1}{2} \left( \sum_{j=1}^{r} \left| \frac{1-\varepsilon_j}{2^m} \right| + \sum_{j=r+1}^{2^n} \frac{\varepsilon_j}{2^m} \right) \leq \frac{1}{2} \left( \frac{r}{2^m} + \frac{2^n - r}{2^m} \right) = \frac{1}{2}\frac{1}{2^{m-n}}$$

$$(20)$$

Hence, $\delta(p,q) \in \mathcal{O}\left(\frac{1}{2^{m-n}}\right)$ which completes the proof. $\square$

Of course, this statement is not the most efficient possible, as the construction utilizes only $0$ and $\pi$ relative phases. By using all possible phases, a stronger result can be obtained as we show next, but auxiliary lemmas should be proved.

## A.2 PROOF OF THEOREM 1

We prove several supplementary lemmas first, before the proof of the main statement.

**Lemma 8.** *Let the total quantum system consist of $m+n$ qubits, where the quantum state is prepared using an IQP circuit, specified by diagonal matrix*

$$D(\theta) = \sum_{x \in \{0,1\}^m} \sum_{y \in \{0,1\}^n} e^{i\theta_{x,y}} |x\rangle\langle x| \otimes |y\rangle\langle y|$$

*with the final layer of Hadamard gates omitted. Then, the reduced density matrix $\rho_2$ of the second, $n$-qubit, subsystem is given by*

$$\rho_2 = \frac{1}{2^m} \sum_{k \in \{0,1\}^m} |\psi_k\rangle\langle\psi_k|, \tag{21}$$

*where each state $|\psi_k\rangle$ takes the form*

$$|\psi_k\rangle = \frac{1}{\sqrt{2^n}} \sum_{y \in \{0,1\}^n} e^{i\theta_{k,y}} |y\rangle, \tag{22}$$

*and the phases $\theta_{k,y}$ are trainable parameters of the IQP circuit.*

*Proof.*

$$\rho_{1+2} = D(\theta) H^{\otimes(m+n)} (|0\rangle\langle 0|)^{\otimes(m+n)} H^{\otimes(m+n)} D^\dagger(\theta)$$

$$= \frac{1}{2^{m+n}} \sum_{x,x' \in \{0,1\}^n} \sum_{y,y' \in \{0,1\}^m} e^{i(\theta_{x,y} - \theta_{x',y'})} \left(|x\rangle\langle x'| \otimes |y\rangle\langle y'|\right). \tag{23}$$

Reduced density matrix for the second subsystem:

$$\rho_2 = \sum_{k \in \{0,1\}^n} \frac{1}{2^{m+n}} \sum_{x,x' \in \{0,1\}^m} \sum_{y,y' \in \{0,1\}^n} e^{i(\theta_{x,y} - \theta_{x',y'})} \langle k|x\rangle\langle x'|k\rangle \otimes |y\rangle\langle y'| =$$

$$= \sum_{y,y' \in \{0,1\}^n} \left( \frac{1}{2^{m+n}} \sum_{k \in \{0,1\}^m} e^{i(\theta_{k,y} - \theta_{k,y'})} \right) |y\rangle\langle y'| =$$

$$= \frac{1}{2^m} \sum_{k \in \{0,1\}^m} \underbrace{\left( \frac{1}{\sqrt{2^n}} \sum_{y \in \{0,1\}^n} e^{i\theta_{k,y}} |y\rangle \right)}_{|\psi_k\rangle} \left( \frac{1}{\sqrt{2^n}} \sum_{y' \in \{0,1\}^n} e^{-i\theta_{k,y'}} \langle y'| \right)$$

$$= \frac{1}{2^m} \sum_{k \in \{0,1\}^m} |\psi_k\rangle\langle\psi_k| \tag{24}$$

$\square$

**Lemma 9** (Representing 2-sparse distributions with IQP state). *For any 2-sparse distributions over $\{+,-\}^n$, i.e., $\exists \tilde{s}_1, \tilde{s}_2 \in \{+,-\}^n$, $P(\tilde{s}_1) = |\alpha|^2$, $P(\tilde{s}_2) = |\beta|^2 = 1 - |\alpha|^2$, there exists a choice of parameters $\theta_k$ for the state in Eq. (22) such that*

$$|\psi_k\rangle = \alpha |\tilde{s}_1\rangle + \beta |\tilde{s}_2\rangle. \tag{25}$$

*Proof.* Eq. (22) implies that the IQP-state is always a *UMA* (uniform magnitude) state, i.e., all amplitudes have the same absolute value w.r.t. the computational basis. The proof comes in the three steps.

**Step 1: A single qubit** Let $\tilde{s}_1 = |+\rangle$, $\tilde{s}_2 = |-\rangle$, $p \in [0,1]$ denote the probability to sample $|+\rangle$. Applying Eq. (22), we have the IQP-state (factoring out the global phase):

$$|\psi_k\rangle = \frac{1}{\sqrt{2}}(|0\rangle + e^{i\theta_k}|1\rangle),$$

Now, if we choose the parameter $\theta$ such that

$$p = |\langle +|\psi_k\rangle|^2 = \cos^2(\theta_k/2),$$

which is a surjective map from $[0, 2\pi]$ to $[0,1]$. Next, we extend it to multi-qubit case.

**Step 2: $\tilde{s}_1, \tilde{s}_2$ only differ in the first position** Now we claim: if $\alpha$ and $\beta$ are such as above, then for all strings $|\tilde{s}_1\rangle$ and $|\tilde{s}_2\rangle$ of $X$ eigenstates, the state

$$|\psi\rangle = \alpha |\tilde{s}_1\rangle + \beta |\tilde{s}_2\rangle$$

is UMA.

If $\tilde{s}_1 = \tilde{s}_2$, then the claim is trivial as we can set $p = 1$.

If not, let $j$ be the first position where $(\tilde{s}_1)_j \neq (\tilde{s}_2)_j$. We can assume $j = 1$. If not, we can swap the first and $j$-th qubit - this state will be UMA if and only if $|\psi\rangle$ was UMA.

So without loss of generality, we can write:

$$|\psi\rangle = \alpha|+\rangle|\tilde{s}_1'\rangle + \beta|-\rangle|\tilde{s}_2'\rangle.$$

and it will suffice to prove this is an UMA state. If $\tilde{s}_1' = \tilde{s}_2'$, we are done as $|\psi\rangle$ factorizes, and the tensor product of a UMA state (on the first qubit by previous analysis) and another UMA state is UMA.

**Step 3: Arbitrary** If $\tilde{s}_1' \neq \tilde{s}_2'$, define $U$ as the $(n-1)$-qubit diagonal unitary such that

$$|\tilde{s}_2'\rangle = U|\tilde{s}_1'\rangle.$$

Note that $U$ is a diagonal unitary with $\pm 1$ on the diagonal. Define the Hadamard-controlled-$U$:

$$\text{hc}U = |+\rangle\langle+| \otimes I_{n-1} + |-\rangle\langle-| \otimes U.$$

Note that: (1)

$$\text{hc}U(\alpha|+\rangle|\tilde{s}_1'\rangle + \beta|-\rangle|\tilde{s}_1'\rangle) = |\psi\rangle,$$

and (2) hc$U$ is a block matrix with blocks $(I + U)/2$ on the diagonal and $(I - U)/2$ on the antidiagonal.

Note that in each row, we have only one non-zero element. So this is a permutation matrix. For a permutation matrix $U$, it holds that $|\psi\rangle$ is UMA if and only if $U|\psi\rangle$ is UMA.

So since

$$\alpha|+\rangle|\tilde{s}_1'\rangle + \beta|-\rangle|\tilde{s}_1'\rangle$$

is UMA by previous analysis, so is $|\psi\rangle$. $\qquad\square$

Next, we present the two key probability distribution decomposition lemmas in the language of probability vectors, i.e., vectors $p = (p_j)_j$, $\sum_j p_j = 1$, $p_j \geq 0$, encoding the probability of the measurement of each of the bitstrings (indexed by $j$). We will say a (probability) vector is $k$-sparse if it has at most $k$ non-zero entries.

**Lemma 10.** *Every $N$-dimensional probability vector $p$ can be expressed as a uniform mixture of $N$ 3-sparse probability vectors. That is, there exists a set $\{q(i)\}_i$ of $N$ 3-sparse probability vectors such that*

$$p = \sum_i \frac{1}{N}q(i).$$

*Proof.* We give a constructive proof by showing there exists an allocation matrix $Q \in \mathbb{R}^{N \times N}$ of $p$ such that the column sum $\sum_i Q_{ij} = p_j$ and the row sum $\sum_j Q_{ij} = 1/N$, which is also 3-row-sparse. The 3-sparse probability vectors are the rows of the $Q$, i.e., $q(i)/N$ is the $i$th row of $Q$. We assume $p_1 \leq p_2 \leq \cdots \leq p_N$ w.l.o.g. Let $k$ be the smallest index such that $p_{k+1} \geq 1/N$. Note $k = 1$ if the distribution is uniform and can be as large as $N - 1$, but not $N$.

The general idea is that we first allocate $p_1, \ldots, p_k$ to the first $k$ diagonal entries of $Q$. Since $p_1 \leq p_2 \leq \cdots \leq p_k \leq 1/N$, we have some capacity left on each row, and hence we can further try to spread out $p_{k+1}$ to the first $k$ rows. After $p_{k+1}$ is fully spread out, we proceed with $p_{k+2}$ and so on. Note that whenever the first rows are out of capacity, we will proceed using the remaining $N - k$ rows. The matrix $Q$ can be constructed and verified with the following three steps.

Step 1: Initialization.

$$Q = \begin{bmatrix} Q_0 & 0 \\ 0 & 0 \end{bmatrix}, \quad Q_0 = \begin{bmatrix} p_1 & 0 & \cdots & 0 \\ 0 & p_2 & \cdots & 0 \\ \vdots & \vdots & \ddots & \vdots \\ 0 & 0 & \cdots & p_k \end{bmatrix}$$

Step 2: Iterate over $\ell = k + 1, \ldots, N$. For each iteration, we spread out $p_\ell$ to the $\ell$ column of $Q$ as follows. For $i$ running from 1 to $N$, we set

$$Q_{i\ell} = \min \left\{ \frac{1}{N} - \sum_j Q_{ij}, p_\ell - \sum_\alpha Q_{\alpha\ell} \right\}. \tag{26}$$

Note that, $N^{-1} - \sum_j Q_{ij}$ is the remaining capacity of row $i$ (recall each row sums to $1/N$) and $p_\ell - \sum_{\alpha<i} Q_{\alpha\ell}$ is the residual of $p_\ell$ after spreading it over the first $i - 1$ row. Two indicator functions check if a row is out of capacity or $p_\ell$ is completely spread out.

Step 3: Correctness. After iteration $\ell = k + 1$, we must have at most two nonzero entries in each row since initially, $Q$ has at most one nonzero value per each row, and by Eq. (26), we only modify one entry of each row. Let $r$ be the largest row index such that $Q_{r\ell} \neq 0, \ell = k + 1$. After the next iteration $\ell = k + 2$, we have two cases: (1) if the capacity of row $r$ is used up by $p_{k+1}$, then the first nonzero entry of column $k + 2$ starts at row $r$. In this case, $Q$ has at most two nonzero entries per row; (2) if some capacity of row $r$ is left, i.e., $\sum_j Q_{rj} \leq 1/N$, then first nonzero entry of column $k + 2$ starts at row $r + 1$, giving rise to three nonzero entries for this row. Note that the above argument between iterations $k + 1$ and $k + 2$ holds for any two iterations $\ell$ and $\ell + 1$. Hence, we conclude that after iteration $\ell \geq k + 1$, the following conditions are true:

1. $\sum_i Q_{ij} = p_j$ for $1 \leq j \leq \ell$;

2. There are at most three nonzero entries in each row of $Q$.

Now, proceed with the above argument until $\ell = N$, we must have: (1) $\sum_j Q_{ij} = 1/N$ for $1 \leq i \leq N$; (2) $\sum_i Q_{ij} = p_j$ for $1 \leq j \leq N$; (3) there are at most three nonzero entries in each row of $Q$. □

Next, we show that using twice as many $q$-vectors, we can represent the probability vector $p$ with 2-sparse probabilities. The basic idea is that a 3-sparse vector can always be written as a uniform mixture of two 2-sparse vectors: Let $p$ be a 3-sparse vector with $0 < p_a \leq p_b \leq p_c$, for indices $a, b, c$. Then we define the probability vectors $q_1$ and $q_2$ as follows: $(q_1)_a = 2p_a$, $(q_1)_c = 1 - 2p_a$, $(q_2)_b = 2p_b$, and $(q_2)_c = 1 - 2p_b$. The correctness can be verified by $q_1, q_2$ are 2-sparse, $q_1/2 + q_2/2 = p$, and $p_a \leq 1/2$ and $p_b \leq 1/2$.

**Lemma 11.** *Every $N$-dimensional probability vector $p$ can be expressed as a uniform mixture of $2N$ 2-sparse probability vectors. That is, there exists a set $\{q(i)\}_i$ of $2N$ 2-sparse probability vectors such that*

$$p = \sum_i \frac{1}{2N} q(i).$$

*Proof.* Taking Lemma 5, we know there exists a set $\{q(i)\}_i$ of $N$ 3-sparse probability vectors such that $p = \sum_i \frac{1}{N} q(i)$. Now, for each $q(i)$, if it is 3-sparse but not 2-sparse, we can always split it into two 2-sparse vectors using the observation before the lemma, i.e., $q(i) = q_1(i)/2 + q_2(i)/2$; If

$q(i)$ is 2-sparse, we simply produce two copies of it. This construction satisfies the conditions of the lemma. □

With these lemmas in place, we can now prove the main result summarised in Theorem 1:

*Proof.* By Lemma 8 the reduced density matrix of an IQP circuit with $m$ hidden qubits can be exactly written as sum of $2^m$ pure states density matrices $|\psi_k\rangle\langle\psi_k|$. The output distribution will be a uniform mixture of the distributions obtained from measuring one of the $|\psi_k\rangle$ states in the basis specified by Pauli-X eigenstates. Each $|\psi_k\rangle$ can be expressed as a superposition of strings of Pauli-X eigenstates, and the probability of observing the string $b \in \{0,1\}^n$ is given by $|\langle\tilde{b}|\psi_k\rangle|^2$ with $|\tilde{b}\rangle = H^{\otimes n}|b\rangle$. In Lemma 11 we have shown how we can encode any 2-sparse distribution in $|\psi_k\rangle$. On the other hand, Lemma 11 states that any probability distribution over $\{0,1\}^n$ can be decomposed as a uniform sum of $2^{n+1}$ such 2-sparse distributions. Therefore, by choosing $m = n + 1$, we have sufficient expressive power to represent any target distribution $p$ exactly using an IQP circuit with hidden qubits. □

**Remark 3.** *Given a target distribution $p$ over $\{0,1\}^n$, we can constructively determine the phase parameters $\theta_{k,y}$ to represent $p$ exactly with IQP-QCBM with hidden qubits. First, we express $p$ as a probability vector $p = (p_1, p_2, \ldots, p_N), N = 2^n$. Second, we convert $p$ to a distribution $\pi$ over $\{+1, -1\}^n$ with Hadamard transform, i.e., $\pi = H^{\otimes n}p$ and a bitstring $b \in \{0,1\}^n$ is mapped to a string $\tilde{s} \in \{+1, -1\}^n$ via $\tilde{s}_i = (-1)^{b_i}$. This transformation is necessary since Lemma 10 works in this sample space. Now, with the algorithms shown in the proof of Lemma 10 and Lemma 11, we can decompose $\pi$ into a uniform mixture of $2N$ 2-sparse distributions over $\{+1, -1\}^n$, i.e., there exist for $k \in [1..2N]$, a 2-sparse distribution $q^{(k)}$ supported on two strings $\tilde{s}(k), \tilde{s}'(k) \in \{+1, -1\}^n$ with probabilities $(\delta^{(k)}, 1 - \delta^{(k)})$ such that $\pi = \frac{1}{2N}\sum_{\ell=1}^{2N} q^{(k)}$. We show that each $q^{(k)}$ can be prepared with the IQP circuit: let $\alpha^{(k)} = \sqrt{\delta^{(k)}}, \beta^{(k)} = \sqrt{1 - \delta^{(k)}} e^{i\varphi_k}$, where the relatively phase $\varphi_k$ can be chosen arbitrarily in $[0, 2\pi)$. The following quantum state implements $q^{(k)}$:*

$$|\psi_k\rangle = \alpha^{(k)}|\tilde{s}(k)\rangle + \beta^{(k)}|\tilde{s}'(k)\rangle,$$

*Representing $|\psi_k\rangle$ in computation basis, we have*

$$|\psi_k\rangle = \frac{1}{\sqrt{2^n}} \sum_{y \in \{0,1\}^n} \left( \alpha^{(k)}(-1)^{\sum_i \mathbb{1}(\tilde{s}_i(k)=-1)y_i} + \beta^{(k)}(-1)^{\sum_i \mathbb{1}(\tilde{s}'_i(k)=-1)y_i} \right)|y\rangle.$$

*Then we can specify phases of the model with hidden qubits. Using Eq. (22) we get:*

$$e^{i\theta_{k,y}} = \alpha^{(k)}(-1)^{\sum_i \mathbb{1}(\tilde{s}_i(k)=-1)y_i} + \beta^{(k)}(-1)^{\sum_i \mathbb{1}(\tilde{s}'_i(k)=-1)y_i}$$

*Thus, given the full probability vector $p$, our proof yields a constructive mapping from $p$ to model parameters $\{\theta_{k,y}\}$. However, we need to mention that this construction requires access to the exact probabilities of $p$ and is therefore not directly applicable in practical learning scenarios, where only a limited number of samples are available. In practice, just as for other universal models (RBMs, GANs, diffusion models), one still needs learning algorithms and inductive bias rather than relying on the explicit universal construction.*

## A.3 HIDDEN QUBITS IMPORTANCE

**Lemma 12** (distributions beyond 2-qubit model expressive power). *Any 3-sparse distribution over $\{0,1\}^2$ with exactly 3 non-zero components cannot be represented by a 2-qubit IQP-QCBM.*

*Proof.* It is equivalent to show that any 3-sparse distribution over $\{+, -\}^2$ with exactly 3 non-zero components cannot be represented by a 2-qubit IQP-QCBM with the final Hadamard layer omitted. W.l.o.g., consider a target distribution with $p_{++}, p_{+-}, p_{-+} > 0$ and $p_{--} = 0$. The model produces distributions of the form

$$p_{\sigma_1, \sigma_2} = \frac{1}{16}\left|1 + \sigma_1 e^{i\theta_1} + \sigma_2 e^{i\theta_2} + \sigma_1\sigma_2 e^{i\theta_3}\right|^2, \tag{27}$$

with $\sigma_1, \sigma_2 \in \{+, -\}$. From $p_{--} = 0$ we get

$$0 = \frac{1}{16}\left|1 - e^{i\theta_1} - e^{i\theta_2} + e^{i\theta_3}\right|^2 \Rightarrow e^{i\theta_3} = e^{i\theta_1} + e^{i\theta_2} - 1. \tag{28}$$

Substituting this into Eq. (27) gives, and writing the normalization condition gives $\sum_{\sigma_1,\sigma_2} p_{\sigma_1,\sigma_2} = 1$ is equivalent here to

$$\cos^2\left(\frac{\theta_1 - \theta_2}{2}\right) + \sin^2\left(\frac{\theta_1}{2}\right) + \sin^2\left(\frac{\theta_2}{2}\right) = 1, \tag{29}$$

This yields the conditions

$$\theta_1 - \theta_2 = \pi \,(\text{mod } 2\pi) \quad \text{or} \quad \theta_1 = 0 \,(\text{mod } 2\pi) \quad \text{or} \quad \theta_2 = 0 \,(\text{mod } 2\pi). \tag{30}$$

In each case at least one of $p_{++}, p_{+-}, p_{-+}$ vanishes: if $\theta_1 - \theta_2 = \pi$ then $p_{++} = 0$; if $\theta_1 = 0$ then $p_{-+} = 0$; if $\theta_2 = 0$ then $p_{+-} = 0$. This contradicts the assumption that all three are strictly positive, completing the proof. $\qquad\square$

**Lemma 13.** *Every probability distribution on $\{0,1\}^2$ can be represented by a 2-qubit IQP–QCBM with* one *hidden qubit.*

*Proof.* Similar to proof of Lemma 12 we ignore last layer of Hadamards by switching to distributions over $\{+,-\}^2$ representation. Recall that the distribution over $\{+,-\}^2$ produced by 2-qubit IQP-QCBM (omitting the last Hadamard layer) can be represented by choosing the parameters in:

$$p_{\sigma_1,\sigma_2} = \frac{1}{16}|1 + \sigma_1 e^{i\theta_1} + \sigma_2 e^{i\theta_2} + \sigma_1\sigma_2 e^{i\theta_3}|^2 \tag{31}$$

with $\sigma_1, \sigma_2 \in \{+,-\}$. We split the proof into steps for easier understanding.

*Step 1: Geometric interpretation of model and target distribution.* First, we can simplify the distribution above produced by the model without hidden qubits:

$$\begin{aligned} p_{\sigma_1,\sigma_2} &= \frac{(1 + \sigma_1 e^{i\theta_1} + \sigma_2 e^{i\theta_2} + \sigma_1\sigma_2 e^{i\theta_3})(1 + \sigma_1 e^{-i\theta_1} + \sigma_2 e^{-i\theta_2} + \sigma_1\sigma_2 e^{-i\theta_3})}{16} \\ &= \frac{1}{4}(1 + \sigma_1 x + \sigma_2 y + \sigma_1\sigma_2 z) \end{aligned}$$

where we denote

$$\begin{cases} x &= \frac{1}{2}\big(\cos\theta_1 + \cos(\theta_2 - \theta_3)\big), \\ y &= \frac{1}{2}\big(\cos\theta_2 + \cos(\theta_1 - \theta_3)\big), \\ z &= \frac{1}{2}\big(\cos\theta_3 + \cos(\theta_1 - \theta_2)\big). \end{cases} \tag{32}$$

Normalization is automatic. Non-negativity is equivalent to that $(x, y, z)$ lies in the tetrahedron:

$$T = \{(x,y,z)| -1 + |x+y| \le z \le 1 - |x - y|\} \tag{33}$$

which is specified by the vertices:

$$\{(1,1,1), (1,-1,-1), (-1,1,-1), (-1,-1,1)\} \tag{34}$$

Thus, the interpretation is the following. An arbitrary target probability distribution is a point that belongs to the tetrahedron $T$, but the parametrized model without hidden qubits only reaches a subset of the tetrahedron that satisfies Eq. (32).

*Step 2: Adding one qubit.* To simplify our calculations further, we purposefully restrict even more Eq. (32). Consider $\theta_3 = \theta_1 + \theta_2 + \delta$ with $\delta \in [0, \pi]$, this leads to simple parametrization of the visible-only qubit

$$\begin{cases} x &= au, \\ y &= av, \qquad a \in [0,1], \qquad u, v \in [-1,1]. \\ z &= uv, \end{cases} \tag{35}$$

where the following notations used:

$$a := \cos(\delta/2) \in [0,1], \quad u := \cos\big(\theta_1 + \tfrac{\delta}{2}\big), \quad v := \cos\big(\theta_2 + \tfrac{\delta}{2}\big) \in [-1,1]. \tag{36}$$

With one hidden qubit, we roughly speaking average two circuits with visible qubits only with independent parameters, which follows directly from Lemma 8, under simplification above it can be written as:

$$\begin{cases} x & = \frac{1}{2}(au + \tilde{a}\tilde{u}), \\ y & = \frac{1}{2}(av + \tilde{a}\tilde{v}), \qquad a, \tilde{a} \in [0,1], \qquad u, v, \tilde{u}, \tilde{v} \in [-1,1]. \\ z & = \frac{1}{2}(uv + \tilde{u}\tilde{v}), \end{cases} \tag{37}$$

*Step 3: Hitting an arbitrary target* $(x, y, z) \in T$. By symmetry, we may assume $x \geq y \geq 0$.

*Case A:* $x > \frac{1}{2}$. Set $a = 1$ and $u = \tilde{v} = 1$. Then

$$\tilde{u} = \frac{2x - 1}{\tilde{a}}, \qquad v = 2y - \tilde{a}, \qquad \tilde{a} \in [2x - 1, 1], \tag{38}$$

and

$$z(\tilde{a}) = \frac{2y - \tilde{a}}{2} + \frac{2x - 1}{2\tilde{a}}, \quad z(1) = x + y - 1, \quad z(2x - 1) = 1 - (x - y). \tag{39}$$

By continuity, $z$ sweeps the full band $[x + y - 1, \ 1 - (x - y)]$.

*Case B:* $x \leq \frac{1}{2}$. Fix

$$a := 1 - 2x, \quad \tilde{a} := 1, \quad u := -1, \quad \tilde{u} := 1, \tag{40}$$

let $v \in [-1, 1]$ be free, and set $\tilde{v} := 2y - (1 - 2x)v$. Then

$$x = \frac{1}{2}(au + \tilde{a}\tilde{u}) = \frac{1}{2}\big(-(1 - 2x) + 1\big) = x, \quad y = \frac{1}{2}(av + \tilde{a}\tilde{v}) = y, \tag{41}$$

and

$$z = \frac{1}{2}(uv + \tilde{u}\tilde{v}) = y - (1 - x)v. \tag{42}$$

As $v$ ranges over $[-1, 1]$, we obtain

$$z \in [y - (1 - x), \ y + (1 - x)] = [x + y - 1, \ 1 - (x - y)]. \tag{43}$$

Feasibility: $u, \tilde{u} \in [-1, 1]$, and since $0 \leq y \leq x \leq \frac{1}{2}$ we have $2y \pm (1 - 2x) \in [-1, 1]$, so $\tilde{v} \in [-1, 1]$ for all $v \in [-1, 1]$.

Combining Cases A and B (and using sign flips/swaps for other quadrants) shows that, for any fixed $(x, y)$, the reachable $z$ coincides with the entire tetrahedron interval $[-1 + |x + y|, \ 1 - |x - y|]$.

$\square$

### A.4 PROOF OF LEMMA 3

*Proof.* Fix any two-bit distribution $\pi$ that *cannot* be realized by a 2-qubit IQP-QCBM without hidden qubits (e.g., any 3-sparse distribution on $\{0,1\}^2$ with exactly three nonzero entries). Define the $n$-bit target

$$p(x_1, \ldots, x_n) = \pi(x_1, x_2) \prod_{k=3}^{n} \mathbf{1}\{x_k = 0\}. \tag{44}$$

*(1) One hidden qubit suffices.* Build an $(n+1)$-qubit IQP-QCBM by taking the 3-qubit construction on qubits (2 visible + 1 hidden) and adjoining $n - 2$ extra visible qubits. By Lemma 13, the 3-qubit construction is universal over 2-bit distributions, so there exists a choice of parameters that implements $\pi$ on the first two visible qubits while acting as the identity on the other $n - 2$ visible qubits; hence, those extra qubits remain deterministically in $|0\rangle$. Therefore, the visible marginal equals $p$.

*(2) Separation for visible-only models.* Suppose, for contradiction, that there exists a parameter setting $\theta$ for an $n$-qubit IQP-QCBM without hidden qubits such that it exactly represents $p(x)$. Because $p(x) = 0$ whenever some $x_k = 1$ with $k \geq 3$, the final state must lie in the span of $\{|x_1 x_2 0 \cdots 0\rangle\}$, and hence factorizes as

$$|\psi\rangle = |\psi_{1,2}\rangle \otimes |0\rangle^{\otimes(n-2)}. \tag{45}$$

Consequently, the output distribution factorizes as

$$q_\theta(x) = r_\theta(x_1, x_2) \prod_{k=3}^{n} \mathbf{1}\{x_k = 0\}, \tag{46}$$

where $r_\theta$ is the distribution produced by some 2-qubit visible-only model. Since $q_\theta = p$, we must have $r_\theta = \pi$, i.e., $\mathrm{TVD}(r_\theta, \pi) = 0$. This contradicts Lemma 12 (which asserts that $\pi$ is not realizable by any 2-qubit visible-only IQP-QCBM).

Therefore, no visible-only $n$-qubit model can realize $p$ exactly. By compactness of the parameter space and continuity of $\theta \mapsto q_\theta$, the minimum of $\mathrm{TVD}(p, q_\theta)$ over $\theta$ is attained and is strictly positive, yielding the claimed constant $C > 0$. $\qquad\square$

### A.5 PROOF OF REMARK 1

*Proof.* The key idea is to express the quantum expectation $\langle Z_\alpha \rangle_{q_\theta}$ as a classical expectation over uniformly random bitstrings, which can be estimated efficiently using Monte Carlo sampling similar to IQP-QCBMs without hidden qubits.

Due to the structure of IQP-QCBM, the expectation $\langle Z_\alpha \rangle_{q_\theta}$ can be re-expressed as an expectation over uniformly random bitstrings:

$$\langle Z_\alpha \rangle_{q_\theta} = \mathbb{E}_{y \sim U_m} \mathbb{E}_{z \sim U_n} \left[ \cos\left( \sum_j \theta_j (-1)^{g_j^{(y)} \cdot y \oplus g_j^{(z)} \cdot z} \left(1 - (-1)^{z \cdot \alpha}\right) \right) \right] \tag{47}$$

Here, $y$ and $z$ are sampled uniformly at random. The vectors $g_j^{(y)}$ and $g_j^{(z)}$ denote the $g_j$ acting on hidden and visible qubits, respectively.

This expectation can be approximated by randomly sampling pairs $(y_k, z_i)$ and computing:

$$\widehat{\langle Z_\alpha \rangle}_{q_\theta} = \frac{1}{|Y||Z|} \sum_{k=1}^{|Y|} \sum_{i=1}^{|Z|} \cos\left( \sum_j \theta_j (-1)^{g_j^{(y)} \cdot y_k \oplus g_j^{(z)} \cdot z_i} \left(1 - (-1)^{z_i \cdot \alpha}\right) \right) \tag{48}$$

By construction, the estimator is unbiased:

$$\mathbb{E}_{Y,Z}\left[ \widehat{\langle Z_\alpha \rangle}_{q_\theta} \right] = \langle Z_\alpha \rangle_{q_\theta} \tag{49}$$

To achieve an additive error $\varepsilon$, it suffices to take

$$|Y||Z| \in \mathcal{O}\left( \frac{1}{\varepsilon^2} \right) \tag{50}$$

samples. Setting $\varepsilon^2 \in \mathcal{O}(\mathrm{poly}(n^{-1}, m^{-1}))$ requires only a polynomial number of samples in $n$ and $m$. Thus, even in the presence of hidden qubits, the expectation value $\langle Z_\alpha \rangle_{q_\theta}$ can be estimated classically to polynomial accuracy using a Monte Carlo method with polynomial runtime.

Additionally, we note that the above rough sample complexity can be refined. Simply applying McDiarmid's inequality to $\widehat{\langle Z_\alpha \rangle}_{q_\theta}$ gives

$$\Pr\left( |\langle Z_\alpha \rangle_{q_\theta} - \widehat{\langle Z_\alpha \rangle}_{q_\theta}| \geq \varepsilon \right) \leq 2 \exp\left( -\frac{2\varepsilon^2}{|Y||Z|n(8\pi \mathrm{poly}(n+m)/|Y||Z|)^2} \right) \tag{51}$$

$$= 2 \exp\left( -\frac{\varepsilon^2 |Y||Z|}{32\pi^2 \mathrm{poly}(n+m)} \right) \tag{52}$$

Thus, to guarantee with probability at most $\delta$ with a deviation $\varepsilon$, a sufficient condition is

$$|Y||Z| \geq 32\pi^2 \mathrm{poly}(n+m) \frac{1}{\varepsilon^2} \log \frac{2}{\delta} \tag{53}$$

$$\square$$

### A.6 PROOF OF LEMMA 4

*Proof.* Let the target distribution be the uniform distribution:

$$p(x) = \frac{1}{2^n}, \quad \forall x \in \{0,1\}^n. \tag{54}$$

Consider an IQP-QCBM circuit with generators $g_j = (0, \ldots, 0, 1, 0, \ldots, 0)$ (a single-qubit $Z$ term). For parameters $\theta' = (\pi/8, \pi/4, \ldots, \pi/4)$, the circuit outputs:

$$q_{\theta'}(x) = \begin{cases} \dfrac{\cos^2(\pi/8)}{2^{n-1}}, & x_1 = 0, \\ \dfrac{\sin^2(\pi/8)}{2^{n-1}}, & x_1 = 1. \end{cases} \tag{55}$$

The TV distance simplifies to:

$$\text{TV}(p, q_{\theta'}) = \frac{1}{2} \sum_{x \in \{0,1\}^n} |p(x) - q_{\theta'}(x)| = \frac{1}{2}\left( 2^{n-1}\left| \frac{\cos^2\left(\frac{\pi}{8}\right)}{2^{n-1}} - \frac{1}{2^n} \right| + 2^{n-1}\left| \frac{\sin^2\left(\frac{\pi}{8}\right)}{2^{n-1}} - \frac{1}{2^n} \right| \right)$$

$$= \frac{1}{4}\left( |2\cos^2(\frac{\pi}{8}) - 1| + |2\sin^2(\frac{\pi}{8}) - 1| \right) = \frac{2}{4}\cos\left(\frac{\pi}{4}\right) = \frac{\sqrt{2}}{4} \tag{56}$$

This is a constant independent of $n$.

The Fourier characteristic for the target and model distribution can be written as follows:

$$\phi_p(\alpha) = \delta_{\alpha,0}, \qquad \phi_{q_\theta}(\alpha) = \prod_{j:\alpha_j=1} \cos(2\theta_j). \tag{57}$$

At $\theta'$:

$$\phi_{q_{\theta'}}(\alpha) = \begin{cases} 1, & \alpha = (0, \ldots, 0), \\ \frac{\sqrt{2}}{2}, & \alpha = (1, 0, \ldots, 0), \\ 0, & \text{otherwise}. \end{cases} \tag{58}$$

Thus, the only nonzero discrepancy occurs at $\alpha^\star = (1, 0, \ldots, 0)$:

$$|\phi_p(\alpha) - \phi_{q_{\theta'}}(\alpha)| = \begin{cases} \frac{\sqrt{2}}{2}, & \alpha = \alpha^\star, \\ 0, & \text{otherwise}. \end{cases} \tag{59}$$

The partial derivatives of model distribution w.r.t. parameter $\theta_k$:

$$\frac{\partial}{\partial \theta_k} \phi_{q_{\theta'}}(\alpha) = \begin{cases} -2\sin(2\theta_k) \prod_{j \neq k, j:\alpha_j=1} \cos(2\theta_j) & \text{if } \alpha_k = 1 \\ 0 & \text{otherwise} \end{cases} \tag{60}$$

For any bounded characteristic kernel $k$ with spectral measure $G(\alpha)$:

$$\text{MMD}_k^2(p, q_{\theta'}) = \sum_{\alpha \in \{0,1\}^n} |\phi_p(\alpha) - \phi_{q_{\theta'}}(\alpha)|^2 G(\alpha) = \tfrac{1}{2} G(\alpha^\star). \tag{61}$$

Gradient structure:

$$\left\| \nabla_\theta \text{MMD}_{k_\sigma}^2(p, q_{\theta'}) \right\| = \sqrt{\sum_{j=1}^n \left( \frac{\partial}{\partial \theta_j} \text{MMD}_{k_\sigma}^2(p, q_{\theta'}) \right)^2}$$

$$= \sqrt{\sum_{j=1}^n \left( \sum_{\alpha \in \{0,1\}^n} 2\left| \phi_p(\alpha) - \phi_{q_{\theta'}}(\alpha) \right| G(\alpha) \frac{\partial}{\partial \theta_j} \phi_{q_\theta}(\alpha) \Big|_{\theta=\theta'} \right)^2}$$

$$= 2\left| \phi_p(\alpha^\star) - \phi_{q_{\theta'}}(\alpha^\star) \right| G(\alpha^\star) \left| \frac{\partial}{\partial \theta_1} \phi_{q_\theta}(\alpha^\star) \Big|_{\theta=\theta'} \right| = 2G(\alpha^\star) \tag{62}$$

- *Gaussian kernel.* The spectral measure admits:

$$G_\sigma(\alpha) = p_\sigma^{|\alpha|}(1-p_\sigma)^{n-|\alpha|}, \quad p_\sigma = \frac{1-e^{-1/(2\sigma)}}{2} < \frac{1}{2}.$$

This gives us an expression for MMD and its gradient norm:

$$\mathrm{MMD}_{k_\sigma}^2(p, q_{\theta'}) = \frac{1}{2}p_\sigma(1-p_\sigma)^{n-1} \in \mathcal{O}(2^{-n}),$$

$$\left\| \nabla_\theta \mathrm{MMD}_{k_\sigma}^2(p, q_{\theta'}) \right\| = 2p_\sigma(1-p_\sigma)^{n-1} \in \mathcal{O}(2^{-n}).$$

- *Smart kernel choice.* Define a kernel with spectral measure concentrated on $\alpha^\star$:

$$G_\kappa(\alpha) = \begin{cases} 1-\varepsilon, & \alpha = \alpha^\star, \\ \dfrac{\varepsilon}{2^n-1}, & \text{otherwise,} \end{cases} \quad 0 < \varepsilon \ll 1.$$

This kernel is characteristic since $G_\kappa(\alpha) > 0$ for all $\alpha$. For MMD and its gradient norm, we have:

$$\mathrm{MMD}_\kappa^2(p, q_{\theta'}) = \frac{1}{2}(1-\varepsilon),$$

$$\left\| \nabla_\theta \mathrm{MMD}_\kappa^2(p, q_{\theta'}) \right\| = 2(1-\varepsilon).$$

Thus, setting $C = \frac{1}{2}(1-\varepsilon)$ gives the desired constant lower bounds, which finishes the proof.

$\square$

### A.7 Proof of Lemma 6

*Proof.* We prove both directions separately.

($\Leftarrow$) Suppose $q_t \xrightarrow{\mathrm{d}} p$. Since $\{0,1\}^n$ is finite, weak convergence is equivalent to convergence in total variation:

$$\lim_{t\to\infty} \mathrm{TV}(q_t, p) = \frac{1}{2}\sum_{x\in\{0,1\}^n} |q_t(x) - p(x)| = 0. \tag{63}$$

By Parseval's identity for the Fourier basis on $\{0,1\}^n$ and Cauchy-Schwarz inequality, we have:

$$\sum_{\alpha\in\{0,1\}^n} |\phi_p(\alpha) - \phi_{q_t}(\alpha)|^2 = 2^n \sum_{x\in\{0,1\}^n} |p(x) - q_t(x)|^2 \quad \leq 2^n \left(\sum_x |p(x) - q_t(x)|\right)^2$$

$$= 2^n \cdot 4\mathrm{TV}(p, q_t)^2 \tag{64}$$

Therefore,

$$0 \leq \lim_{t\to\infty} \mathcal{L}(q_t, p) \leq \lim_{t\to\infty}\left(4\cdot 2^n \mathrm{TV}(q_t, p)^2 \max_{\gamma,\alpha} G_\gamma(\alpha)\right) = 0 \tag{65}$$

($\Rightarrow$) Conversely, suppose $\lim_{t\to\infty}\mathcal{L}(q_t, p) = 0$.

By definition,

$$\mathcal{L}(q_t, p) = \max_\gamma \sum_{\alpha\in\{0,1\}^n} G_\gamma(\alpha)|\phi_{q_t}(\alpha) - \phi_p(\alpha)|^2. \tag{66}$$

Since each $G_\gamma$ has full support, there exists a constant $c > 0$ such that $G_\gamma(\alpha) \geq c$ for all $\gamma$ and all $\alpha$. Thus,

$$\mathcal{L}(q_t, p) \geq c \sum_{\alpha\in\{0,1\}^n} |\phi_{q_t}(\alpha) - \phi_p(\alpha)|^2 = 2^n c \sum_{x\in\{0,1\}^n} |q_t(x) - p(x)|^2 \to 0 \tag{67}$$

where the last equality follows from Parseval's identity.

Applying Cauchy-Schwarz inequality, we obtain

$$\text{TV}(q_t, p) = \frac{1}{2} \sum_{x \in \{0,1\}^n} |q_t(x) - p(x)| \leq \frac{1}{2} \sqrt{2^n \sum_x |q_t(x) - p(x)|^2} \to 0 \qquad (68)$$

Hence, $\text{TV}(q_t, p) \to 0$, which is equivalent to $q_t \xrightarrow{\text{d}} p$ since $\{0,1\}^n$ is finite. $\qquad\square$

### A.8 PROOF OF LEMMA 7

*Proof.* Construct disjoint sets $A, B \subseteq \{0,1\}^n$ of size $m = Cn2^{n/2}$ with $C = 5\ln 2$ using the probabilistic method.

**Step 1: Constructing $A$ with small Fourier coefficients.**
Sample $A \subseteq \{0,1\}^n$ uniformly at random with $|A| = m$. For any fixed $\alpha \neq 0$, define:

$$\phi_p(\alpha) = \mathbb{E}_{x \sim \mathcal{U}_A}[(-1)^{\alpha \cdot x}] = \frac{1}{m} \sum_{x \in A} (-1)^{\alpha \cdot x} \qquad (69)$$

The terms $(-1)^{\alpha \cdot x}$ are bounded in $[-1, 1]$. By Hoeffding's inequality:

$$\Pr\left(|\phi_p(\alpha)| \geq \varepsilon\right) \leq 2\exp\left(-\frac{\varepsilon^2 m}{2}\right) \qquad (70)$$

A union bound over $2^n - 1 < 2^n$ non-zero $\alpha$ gives:

$$\Pr\left(\exists \alpha \neq 0 : |\phi_p(\alpha)| \geq \varepsilon\right) \leq 2^{n+1} \exp\left(-\frac{\varepsilon^2 m}{2}\right) \qquad (71)$$

Substitute $\varepsilon = 2^{-n/4}$ and $m = Cn2^{n/2}$ with $C = 5\ln 2$:

$$\exp\left(-\frac{\varepsilon^2 m}{2}\right) = \exp\left(-\frac{1}{2}(2^{-n/4})^2 Cn2^{n/2}\right) = \exp\left(-\frac{1}{2}2^{-n/2}Cn2^{n/2}\right) = \exp\left(-\frac{Cn}{2}\right)$$

Thus:

$$2^{n+1}\exp\left(-\frac{Cn}{2}\right) = 2^{1-\frac{3n}{2}} \leq 2^{-n} \quad \text{for } n \geq 2 \qquad (72)$$

So with probability at least $1 - 2^{-n}$:

$$\forall \alpha \neq 0, \quad |\phi_p(\alpha)| \leq 2^{-n/4} \qquad (73)$$

**Step 2: Constructing $B$ disjoint from $A$ with small Fourier coefficients.**
Set $N = 2^n$. Let $A^c = \{0,1\}^n \setminus A$ and sample $B \subseteq A^c$ uniformly with $|B| = m$. For any fixed $\alpha \neq 0$, define:

$$\phi_q(\alpha) = \mathbb{E}_{x \sim \mathcal{U}_B}[(-1)^{\alpha \cdot x}] \qquad (74)$$

Since the total Fourier coefficient over $\{0,1\}^n$ is zero for $\alpha \neq 0$:

$$0 = \frac{m}{N}\phi_p(\alpha) + \frac{N-m}{N}\mu(\alpha), \qquad (75)$$

where

$$\mu(\alpha) = \mathbb{E}_{x \sim \mathcal{U}_{A^c}}[(-1)^{\alpha \cdot x}] \qquad (76)$$

Solving for $\mu(\alpha)$:

$$\mu(\alpha) = -\frac{m}{N-m}\phi_p(\alpha) \qquad (77)$$

Conditioned on $|\phi_p(\alpha)| \leq 2^{-n/4}$ for all $\alpha \neq 0$ we get:

$$|\mu(\alpha)| \leq \frac{m}{N-m}2^{-n/4} \leq \frac{2m}{N}2^{-n/4} = 2Cn2^{-3n/4} \qquad (78)$$

By the triangle inequality:

$$|\phi_q(\alpha)| \leq |\phi_q(\alpha) - \mu(\alpha)| + |\mu(\alpha)| \leq |\phi_q(\alpha) - \mu(\alpha)| + 2Cn2^{-3n/4}$$

Apply Hoeffding's inequality to $|\phi_q(\alpha) - \mu(\alpha)|$:

$$\Pr\left(|\phi_q(\alpha) - \mu(\alpha)| \geq \delta\right) \leq 2\exp\left(-\frac{\delta^2 m}{2}\right) \tag{79}$$

Set $\delta = 2^{-n/4}$. Then:

$$\Pr\left(|\phi_q(\alpha) - \mu(\alpha)| \geq 2^{-n/4}\right) \leq 2\exp\left(-\frac{2^{-n/2}Cn2^{n/2}}{2}\right) = 2\exp\left(-\frac{Cn}{2}\right)$$

With $C = 5\ln 2$:

$$2\exp\left(-\frac{5n\ln 2}{2}\right) = 2^{1-\frac{5n}{2}} \tag{80}$$

Union bound over $\alpha \neq 0$: for $n \geq 2$,

$$\Pr\left(\exists \alpha \neq 0 : |\phi_q(\alpha) - \mu(\alpha)| \geq 2^{-n/4}\right) \leq 2^{1-\frac{3n}{2}} \leq 2^{-n} \tag{81}$$

Thus, with high probability:

$$|\phi_q(\alpha) - \mu(\alpha)| \leq 2^{-n/4}, \quad \forall \alpha \neq 0 \tag{82}$$

Then for large $n$:

$$|\phi_q(\alpha)| \leq 2^{-n/4} + 2Cn2^{-3n/4} \leq 2^{-n/4}(1 + 2Cn2^{-n/2}) \leq 2^{1-n/4}$$

**Step 3: Existence and disjoint support.**
Step 1 succeeds with probability $\geq 1 - 2^{-n}$, and Step 2 with conditional probability $\geq 1 - 2^{-n}$. So both succeed with:

$$(1 - 2^{-n})^2 \geq 1 - 2^{-n+1} > 0 \quad \text{for } n > 1 \tag{83}$$

Such sets $A$ and $B$ exist. Define:

$$p = \mathcal{U}_A, \quad q = \mathcal{U}_B \tag{84}$$

Since $A \cap B = \emptyset$:

$$\mathrm{TV}(p, q) = 1 \tag{85}$$

and for all $\alpha \neq 0$:

$$|\phi_p(\alpha)| \leq 2^{-n/4}, \quad |\phi_q(\alpha)| \leq 2^{1-n/4} \tag{86}$$

$\square$

## A.9 NUMERICAL EXPERIMENTS DETAILS

Here we describe the detailed settings for the numerical experiments with the synthetic parity check datasets. The parity-check matrices behind the ground truth distributions are

$$H_{12} = \begin{bmatrix} 111111110000 \\ 000011111111 \end{bmatrix} \tag{87}$$

$$H_{14} = \begin{bmatrix} 11111111100000 \\ 00000111111111 \end{bmatrix} \tag{88}$$

$$H_{16} = \begin{bmatrix} 1111111111000000 \\ 0000001111111111 \end{bmatrix} \tag{89}$$

respectively. To prepare the training and testing datasets, bit strings $x$ are uniformly randomly sampled from $\{0, 1\}^n$, accepted into the dataset if $Hx = 0$, and rejected if $Hx = 1$.

All experiment runs share the same hyperparameters: We use 2000 training data. To compute the MMD losses during training, we use batches of 1000 frequencies. For each frequency, we compute the quantum part of the MMD losses (Z expectation values) using 1000 measurements, and the data

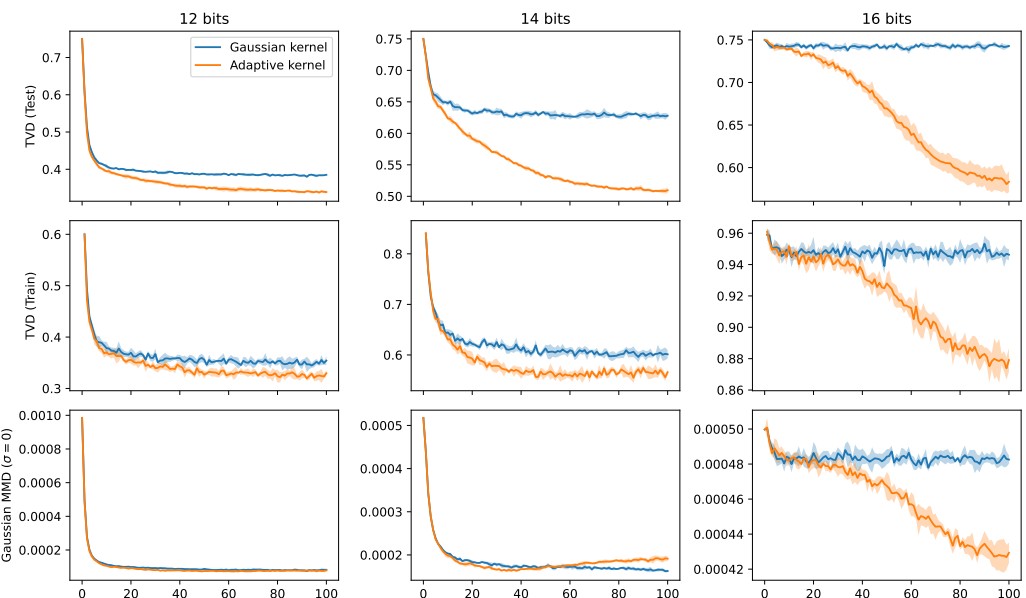

Figure 3: Learning curves for (mean ± standard deviation over 5 runs) on synthetic parity-check datasets of dimension 12, 14 and 16. The first row shows the total variation distance computed between the true generated distribution and ground truth data distribution (same as Figure 2 in the main text). The second row shows the total variation distance computed between the empirical distribution of a batch of 1000 generated data and the empirical distribution of the 2000 training data. The third row shows the empirical MMD with respect to the 0-bandwidth Gaussian kernel, based on 1000 generated data and 2000 training data.

part of the MMD losses using all training data. For updating the IQP generator, we use the Adam optimizer with learning rate 0.001 and $\beta_1 = 0.9, \beta_2 = 0.999$. For updating the adaptive kernel, we use the Adam optimizer with learning rate 0.0001 (decay by 10% every 500 iterations) and $\beta_1 = 0.9, \beta_2 = 0.999$. For the adaptive training runs, we update the generator and kernel at a rate of $1 : 1$.

Now we show how FVSBN, the model we use to implement the parameterized spectral measure, can be initialized to recover a Gaussian spectral measure with bandwidth $\sigma$, we set $W = 0$, and $\forall 1 \leq i \leq n, b_i = \log(p - \epsilon) - \log(1 - p - \epsilon)$ where $p = (1 - \exp(-1/(2\sigma^2)))/2$. Note that for the special case of $\sigma = 0$, we simply take $b_i = 0$.

We show the detailed experimental results in Fig. 3. We observe that for the 12- and 14-bit cases, both Gaussian ($\sigma = 0$) and Adaptive ($\sigma = 0$) methods converge to a significantly lower total variation distance (TVD) on the training set, while yielding substantially higher TVD on the test set, indicating overfitting. For the 16-bit case, the Gaussian ($\sigma = 0$) method shows a flat learning curve since (1) the 2000 training points are quite sparse in $\{0, 1\}^{16}$ and (2) its spectral measure is uniform, which fails to reflect the difference between the generator and the training set. The Adaptive ($\sigma = 0$) method is initialized from the Gaussian and overfits to the training data as it converges to a much lower training TVD. As for the MMD loss, we show the Gaussian MMD ($\sigma = 0$) values for these two methods in the bottom row of Fig. 3. Both Gaussian ($\sigma = 0$) and Adaptive ($\sigma = 0$) converge to near-zero values.

