# OpenReview forum: "Universality and kernel-adaptive training for classically trained, quantum-deployed generative models"
_ICLR.cc/2026/Conference — Submitted to ICLR 2026_

### Official Review · Reviewer_47J7 · 2025-10-26

**Soundness:** 3
**Presentation:** 3
**Contribution:** 2
**Rating:** 6
**Confidence:** 3

**Summary:**

This work presents a well-structured and technically solid QML study on enhancing the Instantaneous Quantum Polynomial (IQP) Quantum Circuit Born Machine (QCBM). The authors identify two major limitations of existing IQP-QCBMs including non-universality and training instability under a fixed Gaussian kernel MMD loss. Then, they theoretically prove that adding $n+1$ ancillary qubits enables universality for an n-qubit generator, and introduce kernel-adaptive MMD loss to improves training stability under a min-max optimization framework.

**Strengths:**

1. it is well-written and structured, making it easy for readers to follow the main arguments and technical developments.
2. the authors provides rigorous proofs of addressing two main questions that are the universality of IQP-QCBM when extended with ancillary qubits and the theoretical advantage of kernel-adaptive MMD training in terms of discriminative power and convergence guarantees.

**Weaknesses:**

the paper would benefit from more comprehensive numerical experiments to substantiate the claims and demonstrate the empirical behavior of the proposed approach under different settings.

**Questions:**

1. Theorem 1 indicates that $m = n + 1$ ancilla qubits are theoretically required to represent an arbitrary probability distribution over ${0,1}^n$ by tracing out the ancilla subsystem. However, in the empirical example shown in Fig. 1, the model achieves reasonable performance with only a single ancilla qubit. This observation raises several important questions. Does the representation of most practical probability distributions indeed require fewer than $m = n + 1$ ancilla qubits? Is the required number of ancilla qubits dependent on the structure or complexity of the target distribution? Furthermore, how can one systematically determine the minimal number of ancilla qubits necessary to faithfully approximate a given probability distribution?

2. Given that the enhanced IQP-QCBM serves as a quantum generative model, it would be valuable to further assess its performance on more realistic and complex datasets beyond the synthetic parity-check case. How does the proposed model perform on more practical benchmarks such as structured classical datasets or quantum states?

---

> ### Author Response · Authors · 2025-11-21
>
> We thank the reviewer for carefully reading the manuscript and for all the comments and questions. Please find our responses below.
>
> **W and Q2.** We agree it would be interesting to benchmark the model on more realistic and structured datasets. However, in this work, we focus on what we find to be the more immediate, fundamental questions. Allow us to argue this. IQP-type models are considered a ground-breaking family because they are both classically trainable and are known to show quantum advantages in sampling. So, in principle, they would allow us, for the first time, to obtain some ideas about when quantum computers could do well in ML applications before large devices are available, which would be a clear game-changer, as in ML empirical performance has primacy. Prior to our work, however, it was not known whether these models are universal, and it was not clear how limited they were. It could have been the case that they capture only irrelevant distributions, in which case all this potential would be lost. Further, only one loss function was known to be computable, limiting how informative the tests would be.
> In this work, we tackle what is to us clearly the key question, and show that by adapting the model, we can reach universality with little cost, and that we can even significantly generalize the spectrum of loss functions that can be computed. In this paper, we analyze this aspect. Second to this question, we find the questions of performance on real, general datasets (which now makes sense since the generalized model is universal), questions of trainability, and questions of robustness under experimental noise, all of which are very important but not the focus of this work.
>
> **Q1.** In Theorem 1, we prove that $ n+1$ hidden qubits suffice in the worst-case scenario to express any target distribution exactly. However, the required number of hidden qubits depends on the structure of the target distribution and can be much smaller than $n+1$. In the paper, we show that there exist two-bit distributions that cannot be expressed without hidden qubits but can be expressed with just one additional qubit, rather than three extra qubits. In the revised version, we generalize this: there exists an
> $n$-bit distribution that requires only one hidden qubit to be expressed exactly. Systematically determining the minimal ancilla count for an arbitrary distribution is a nontrivial open problem; however, in practice, we view the number of hidden qubits as a model-capacity hyperparameter to be chosen based on validation performance.

---

> > ### Comment · Reviewer_47J7 · 2025-11-27
> >
> > Thank you for the response.
> >
> > W and Q2. I still believe providing experiments on real, rather than synthetic, datasets is still necessary to convincingly demonstrate that the IQP-type model is not merely a conceptual toy but a practically relevant approach.
> >
> > Q1. Could you please provide the details of the what's kind of distribution that only requires one hidden qubit to be expressed?

---

> > > ### Author Response · Authors · 2025-12-03
> > >
> > > Thank you for the response. Please see our replies below.
> > >
> > > **W and Q2.** We thank the reviewer for the suggestion, but we respectfully disagree that a substantially larger experimental study is necessary to support the main claims of this work.
> > >
> > > First, IQP–QCBM–type models have already been trained and evaluated in prior work (e.g., Recio-Armengol et al., 2025), where they were compared against classical generative models such as restricted Boltzmann machines on several datasets. These studies show that IQP-style generative models can match the performance of standard classical baselines, demonstrating their practical viability. Reproducing a similar empirical evaluation in our paper, in our view, largely repeats the existing works and would not significantly strengthen our main contribution.
> > >
> > > More importantly, prior numerical work compares the standard IQP mode, which is not universal, to classical universal generative models. This limits the interpretability of those empirical results. Therefore, a key open question is how to extend the IQP framework so that it becomes universal, enabling meaningful comparisons with other universal models. **Addressing this gap is one of the core contributions of our paper: we prove that IQP circuits augmented with hidden qubits become universal, thereby resolving this open problem and establishing a rigorous foundation for IQP-based generative modelling.**
> > > In the experiments, although the dataset we used is synthetically generated, it captures structural features that are characteristic of real-world problems: parity-type pass/fail labels naturally arise in communication and signal-processing tasks, in error-correction scenarios, and in models with multi-spin interactions such as certain Ising or spin-glass–like lattice systems.
> > >
> > > Given that our primary contributions are theoretical, we believe that significantly expanding the empirical study, while certainly interesting and valuable as future work, is beyond the intended scope of this paper.
> > >
> > > **Q1.** Thank you for the question. In the revised version of the paper,  we include the detailed description of these distributions in the proof of Lemma 3: for each system size $n$, IQP-QCBM with a single hidden qubit is provably sufficient to represent this family of distributions exactly, while the corresponding model without ancillas cannot represent them. This family is a natural generalization of a 2-qubit distribution that already requires only one ancilla, which is strictly fewer than the sufficient number guaranteed by our general bound (which, for $n=2$ is $3$). This result demonstrates that the bound in Theorem 1 is a worst-case guarantee rather than a typical requirement, which is encouraging for near-term applications.

---

### Official Review · Reviewer_xDCp · 2025-10-31

**Soundness:** 2
**Presentation:** 2
**Contribution:** 2
**Rating:** 4
**Confidence:** 3

**Summary:**

The authors explore improvements to the Instantaneous Quantum Polynomial Quantum Circuit Born Machine, a type of quantum generative model that produces bit-string samples. The authors tackle two main issues: first, the model’s lack of universality, its inability to represent all possible distributions and second, training difficulties arising from using a fixed Gaussian kernel in the MMD loss. They prove that adding n+1 hidden qubits to an n-qubit IQP generator can make it universal, at least theoretically. They also propose a kernel-adaptive training approach, where the kernel is trained adversarially to better guide optimization. Their experiments on a parity-check dataset show improved training stability and better performance, especially as system size increases.

**Strengths:**

S1. The theoretical contribution around proving universality with hidden qubits is a strong and novel result, it clarifies fundamental expressivity limits in a clean, rigorous way.

S2.the adaptive kernel method is a practical and well-motivated innovation, bridging a clear gap in MMD-based quantum model training and showing meaningful empirical gains.

S3. The authors present a direct discussion in of current limitations and their proposed solutions, both in the introduction but also in their method section, e.g. showing that the that MMD with the Gaussian kernel might be ineffective in distinguishing probability
distribution.

S4. For the numerical results the Adaptive kernel shows a very clear, while realistic, improvement over the Gaussian Kernel.

**Weaknesses:**

W1. The universality proof is largely theoretical, with limited insight into whether such hidden-qubit setups are realistic or scalable in current hardware.

W2. The evaluation remains narrow, relying mostly on synthetic data and not demonstrating performance on more complex or real-world distributions, which makes the broader significance of the results a bit speculative.

W3. The theoretical work is on a specific class of quantum generative models, limiting the significance of this work.

W4. The numerical experiments are very limited, limiting again the generalizability and real-world confidence of the theoretical results

W5. The work is missing some good related work comparison, either in the introduction or separate section.

**Questions:**

Q1. Can the method be compared to in a more generalizable way, specifically more existing methods as well as more synthetic datasets? -if not can you please describe thoroughly why only this comparison is reasonable and comment on the possible lack on benchmark tests.

Q2.  Can you comment on the significance of this work in the more general context of generative models e.t.c. e.g. what is this work aiming to achieve/influence in the future.

Q3. Is it possible to make the proof of theorem 1 constructive, explicitly map a target distribution $p$ to specific phase parameters $\theta_{k,y}$?

---

> ### Author Response · Authors · 2025-11-21
>
> We thank the reviewer for the feedback. Please see the responses for all the comments and questions, which were split into several groups.
>
> **W1.** Regarding scalability/realism, we do not think our overhead for universality is prohibitive. To elaborate, we point out two things:
> - From the perspective of the deployment of the model on quantum hardware, in Theorem 1, we proved that at most n+1 hidden qubits are needed in the worst-case scenario. Roughly speaking, our model only requires to double the number of qubit register in the worst-case scenario, and in our opinion, this is realistic/implementable given the progress in NISQ quantum computers. Furthermore, the expressivity and the number of qubits are hyperparameters, and indeed, it may be the case that for a given problem, one can do sufficiently well using fewer than n+1 hidden units. For instance, in the paper, we showcased that there exist two-bit distributions that cannot be expressed without hidden qubits, but can be expressed with just one more qubit, instead of three extra qubits. In the revised version, we generalize it: there exists an $n$-bit distribution that only requires one hidden qubit to express exactly.
> - From the perspective of classical trainability, for the hidden-qubit model, we have results showing that it can be efficiently trained on classical hardware in the same regime as the model with visible-only qubits, so this should not be an issue.
>
> **W2, W4, Q1.**  First of all, we of course agree that the dataset is synthetic. Since our work is the first to introduce methods that allow us to go beyond Gaussian kernels for IQP-QCBM, we believe it is most important to clearly demonstrate the potential performance differences. For the choice of the parity-check data set, our motivation is two-fold:
> - It poses great challenges to the standard Gaussian kernel: it assigns super-exponentially decaying weights to high/low frequencies in the characteristic function of a distribution. However, the parity-check data sets concentrate on these frequencies, which cannot be effectively measured by a Gaussian kernel. This is a concrete, practical case where we know precisely why Gaussian fails and the adaptive one can outperform.
> - It represents characteristics of many real-world data sets, e.g., in communication/signal processing/error-correction, where the pass/fail label is typically the result of some parity check.
>
> Furthermore, we expect that there will be many other datasets that, incidentally or for some opaque reason, have features that challenge the Gaussian kernel and are perhaps difficult to identify without actually training the models. We do agree that finding these is important. Extending the empirical evaluation to more complex real-world datasets is an important and complementary direction that we plan to pursue in follow-up work.
>
> **W3, Q2.** Indeed, our paper studies a particular class of generative models, but we argue that it is an extremely important and new class. These IQP-type models are a ground-breaking family because they are both classically trainable and are known to show quantum advantages in sampling. They allow us, for the first time, to gain reliable insights into what quantum computers could do well in ML applications before large devices are available, which is a clear game-changer. Prior to our work, however, it was not known whether these models are universal, and it was not clear how limited they were. This is in contrast to other quantum architectures such as QBMs, quantum GANs or EVSs, which have been proven universal and capable of advantage in other works. The shortcoming was that none of these models can be trained at scale without a large, currently non-existing quantum computer. With our results, IQP-QCBMs with hidden units have the best of both worlds: an effective way to train while being universal, and furthermore, we show they allow a plethora of metrics with respect to which they can be trained.
>
> **W5.**  We are happy to revise the paper according to the reviewer’s suggestion, if they could be a bit more specific here. To the best of our knowledge, we have consulted the most recent SOTA works on quantum generative models for bitstrings.
>
> **Q3.** Indeed, the answer is yes! Our proof can be made constructive: given the full target distribution, one can explicitly synthesise a hidden-qubit IQP-QCBM whose phase parameters reproduce it (the mapping is not unique, the construction is based on Lemma 8)

---

> > ### Comment · Reviewer_xDCp · 2025-11-25
> >
> > W1 How does the introduction of hidden qubits affect the circuit depth and total gate count? Specifically, does this added complexity introduce too much noise for current NISQ hardware?
> >
> > W2, W4, Q1 Can you briefly mention one specific real-world domain (beyond error correction) where these difficult features are likely to appear?
> >
> > W3, Q2 Thank you for the clarifications.
> >
> > W5. The comment was for a dedicated related-works section. This is not about citing specific papers, but about allocating space in the manuscript for a complete and well-structured discussion. Please refer to related-works sections in other ICLR papers as guidance.
> >
> > Q3. Did you try to make it constructive? Would that be a reasonable improvment for this work?

---

> ### Author Response · Authors · 2025-11-28
>
> **W1** Thank you for the question. In principle, there is no direct connection between the number of hidden qubits and the total gate count. Just as neural networks have several essentially independent structural parameters (number of layers, number of neurons per layer, connectivity, etc.), we introduce the number of hidden neurons as one more, in-principle independent, structural parameter needed for universality. For a fixed circuit depth and a constant number of gates, adding hidden units changes the set of distributions we can represent and, as we show, at large depths, not only changes but strictly enlarges this set. In practice, users will need to choose architectures appropriate for their specific task, which will determine the distributions they can access. From the perspective of NISQ hardware, one could argue that gate counts and circuit depth are more strongly constrained than qubit counts. In that sense, using additional qubits to access a richer family of distributions could even be more NISQ-friendly than relying on larger depths. A detailed analysis of this trade-off is beyond the scope of the present work, however.
>
> **W2, W4, Q1** We agree that this is an important question, but it is also extremely difficult to address for natural generative modelling problems, because the data-generating processes are challenging to characterize mathematically (e.g., what makes an image “dog-like”?). For more specialized cases, however, we can provide explicit examples. One comes from the physical domain, where parity-structure patterns appear in classical spin systems, for instance in Ising models or spin-glass–like lattice models with multi-spin (plaquette) interactions. In the low-temperature regime, configurations are constrained by products of several spins around a loop. In the $\\{0, 1\\}^n$ encoding, these product constraints become XOR/parity checks on the corresponding bits, so the equilibrium distribution at low temperature is concentrated on a subset of bit strings defined by many global parity constraints.
>
> **W5.** Thank you for the clarification and the suggestion! In the revised version, we have created a new section on related works as you suggested. The manuscript has been updated in the system.
>
> **Q3.** Thank you for your comment. We agree that your suggestion is reasonable and improves the clarity of our results. We added it as Remark 3, p. 17, in the updated version. Below, we outline the sketch of this mapping.
>
> Given a target distribution $p$ over $\\{0,1\\}^n$, we can constructively determine the phase parameters $\theta_{k,y}$ to represent $p$ exactly with IQP-QCBM with hidden qubits. First, we express $p$ as a probability vector $p=(p_1, p_2, \ldots, p_N), N=2^n$. Second, we convert $p$ to a distribution $\pi$ over $\\{+1,-1\\}^n$ with Hadamard transform, i.e., $\pi = H^{\otimes n}p$ and  a bitstring $b\in \\{0,1\\}^n$ is mapped to a string $\tilde{s}\in \\{+1,-1\\}^n$ via $\tilde{s}_i = (-1)^{b_i}$. This transformation is necessary since Lemma 10 works in this sample space.
>
> Now, with the algorithms shown in the proof of Lemma 10 and Lemma 11, we can decompose $\pi$ into a uniform mixture of $2N$ 2-sparse distributions over $\\{+1,-1\\}^n$, i.e., there exist for $k \in[1..2N]$, a 2-sparse distribution $q^{(k)}$ supported on two strings $\tilde{s}(k),\tilde{s}'(k)\in \\{+1,-1\\}^n$ with probabilities $(\delta^{(k)}, 1 - \delta^{(k)})$ such that $\pi = \frac{1}{2N}\sum_{\ell=1}^{2N} q^{(k)}$. We show that each $q^{(k)}$ can be prepared with the IQP circuit: let $\alpha^{(k)}=\sqrt{\delta^{(k)}}$, $\beta^{(k)}=\sqrt{1-\delta^{(k)}} e^{i\varphi_k}$, where the relatively phase $\varphi_k$ can be chosen arbitrarily in $[0,2\pi)$. The following quantum state implements $q^{(k)}$:
> $$
> |\psi_k\rangle = \alpha^{(k)}|\tilde{s}(k)\rangle + \beta^{(k)}|\tilde{s}'(k)\rangle
> $$
> Representing $|\psi_k\rangle$ in the computational basis, we can specify phases of the model with hidden qubits. Using Eq.(22) we get:
>
> $$
> e^{i\theta_{k,y}} = \alpha^{(k)}\\,(-1)^{\sum_i y_i\mathbb{1}(\tilde{s}_{i}(k)=-1)} + \beta^{(k)} (-1)^{\sum_i y_i \mathbb{1}(\tilde{s}_i'(k) = -1)}
> $$
>
> Thus, given the full probability vector $p$, our proof yields a constructive mapping from $p$ to model parameters $\\{\theta_{k, y}\\}$. However, we need to mention that this construction requires access to the exact probabilities of $p$ and is therefore not directly applicable in practical learning scenarios, where only a limited number of samples are available. In practice, just as for other universal models (RBMs, GANs, diffusion models), one still needs learning algorithms and inductive bias rather than relying on the explicit universal construction.

---

### Official Review · Reviewer_omgV · 2025-11-01

**Soundness:** 3
**Presentation:** 2
**Contribution:** 3
**Rating:** 6
**Confidence:** 3

**Summary:**

This paper focuses on advancing the IQP QCBM, a quantum generative model for bitstring generation that is classically trainable and quantum-deployable, by addressing two key limitations of existing versions. First, the base n-qubit IQP-QCBM is non-universal, meaning it cannot represent arbitrary probability distributions. The authors prove that adding n+1 hidden qubits enables exact universality, and even a small number of hidden qubits significantly improves approximation accuracy. Second, training with a fixed Gaussian kernel in MMD often causes vanishing gradients or fails to distinguish distinct distributions. The authors propose a kernel-adaptive adversarial training method: a critic network dynamically tunes the kernel’s spectral measure to focus on frequencies where the model and data distributions differ most. Theoretically, they show this adaptive kernel has stronger discriminative power, and MMD convergence under this method guarantees convergence in distribution. Experiments on synthetic parity-check datasets (12 to 16 bits) confirm that kernel-adaptive training outperforms fixed Gaussian kernels in minimizing TV distance between generated and true distributions, with the performance gap widening as qubit count increases. The paper also notes a limitation: MMD-based training (even with adaptive kernels) fails for certain "worst-case" distributions where TVD is maximal but MMD remains exponentially small.

**Strengths:**

Resolves two limitations of IQP-QCBM
Advances "classical training, quantum deployment", critical for scarce quantum hardware
Reusable frameworks (hidden-qubit universality, spectral kernel adaptation) extend to other quantum models
The theory seems rigorous

**Weaknesses:**

Exclusively uses synthetic parity-check datasets
Proves n+1 hidden qubits suffice for universality but no analysis of minimal qubit count
Adaptive kernel’s FVSBN has O(n^2) parameters. no runtime/memory comparisons vs. fixed Gaussian kernels for large n

**Questions:**

how do hardware noise/cross-talk affect universality? Have you tested small-scale on real quantum hardware?
Did you sweep ε or FVSBN initializations?
Experiments show test TVD > training TVD. Did you try regularization?
Can your framework adapt to IQP-QCBM for continuous quantum states (e.g., chemistry)? If yes, what spectral measure modifications are needed?

---

> ### Author Response · Authors · 2025-11-22
>
> We are very grateful to the reviewer for their comments and questions. Please find our responses below.
>
> **W1.**  We agree that the dataset is synthetic. Since our work is the first to introduce methods that go beyond Gaussian kernels for IQP-QCBMs, our priority is to clearly demonstrate potential performance differences. The parity-check dataset was chosen for two reasons:
> -  This is a concrete, practical case where we know precisely why Gaussian fails and the adaptive one can outperform.
> -  It reflects key properties of real-world datasets, e.g., in communication, signal processing, and error correction, where pass/fail labels often arise from parity checks.
>
> We also expect other datasets will, for incidental or opaque reasons, challenge the Gaussian kernel and may only be identifiable by actually training models. Finding such datasets and extending the empirical evaluation to more complex real-world data is an important, complementary direction for future work.
>
> **W2.** This is an interesting open problem that we have thought about extensively. We suspect that the “+1” is an artefact of our approach, though this is not yet proven. We attempted to show tightness or find more efficient general constructions, but without success so far. We did, however, obtain partial progress. In the revised version, we prove that a single hidden qubit already suffices for universality of 2-qubit IQP-QCBMs, so $n +1$ is not tight for all n (though it may still be tight for large
> n). On the other hand, we also show that even modest increases in the number of hidden qubits strictly improve expressivity: there exist distributions over n bits that no IQP-QCBM without hidden qubits can represent, but that become exactly representable after adding just one hidden qubit.
>
> **W3.** First, in our humble opinion, comparing time complexity is not critical because the Gaussian kernel can never reach the target functions (regardless of the time). Second, it is indeed quadratic in the number of qubits. But the operation boils down to matrix multiplication of size n by n, which, from our perspective, is not a computation bottleneck for the moderate size of n. For large n, we can reduce this cost by using a different graph model to represent the spectral measure, e.g., only conditioning on a limited/fixed-sized neighbourhood for each bit.
>
> **Q1.**  Our work is the first to prove the universality of these models and one of the first papers on this model in general. Consequently, we focus on the most fundamental questions and have therefore opted to leave hardware constraints and limitations for future work. In this sense, our universality result is proved for the ideal, noiseless IQP-QCBM family, which is the standard setting for universality theorems in (quantum) machine learning. Noise is a property of a particular hardware implementation. In the most general terms, realistic quantum noise will certainly somewhat decrease universality, as, e.g., Kronecker-delta distributions cannot be represented exactly in the presence of any finite amount of, say, measurement error.
>
> **Q2.** The main motivation for the IQP-QCBM paradigm is that we can obtain insights without training on real hardware, so the most immediate questions lie away from hardware considerations. Hence, our focus is on fundamental questions of expressivity. However, sampling at scale will require a real device, and hardware noise will affect results. Using quantum devices for inference (to generate samples) and systematically studying how hardware noise affects the learned output distributions is an important direction for future work.
>
> **Q3.** Yes! We performed a sweep with $\varepsilon$ in 10^\{-8, -7, -6, -5, …, -1\} and we chose $10^{-6}$ as it produces the most stable training loss curves.
>
> **Q4.** We would respectfully disagree that Figure 4 showcases overfitting. To explain, a smaller TVD on training/test data sets indicates the generator’s distribution is closer to the target, meaning that the smaller TVD values in Figure 4 the better performance. Figure 4 shows exactly the opposite of what you pointed out. On the training set (the middle row of Figure 4), the adaptive training method is similar to or better than the fixed Gaussian kernel, and on the test set (first row), it is much better than the Gaussian, suggesting that the adaptive training is less prone to overfitting.
>
> **Q4-5.** The most straightforward extension to continuous random variables is to encode the real values as bitstrings up to some precision, which requires a huge number of qubits to implement. In this case, the current approach to parameterised spectral measure can be directly used. However, we would like to point out that for modelling continuous probability distributions, it is more natural to use models like Q-GANs (e.g. Expectation Value Samplers).

---

### Official Review · Reviewer_33e5 · 2025-11-03

**Soundness:** 3
**Presentation:** 3
**Contribution:** 3
**Rating:** 6
**Confidence:** 3

**Summary:**

This paper addresses two key limitations of IQP-QCBM (Instantaneous Quantum Polynomial Quantum Circuit Born Machines): (1) lack of universality and (2) training difficulties with fixed Gaussian kernels in MMD-based objectives. The authors prove that adding n+1 hidden qubits makes the n-qubit IQP-QCBM universal, and propose an adversarial kernel-adaptive training method in which the kernel's spectral measure is learned. Experiments on synthetic parity-check datasets demonstrate improved performance with adaptive kernels.

**Strengths:**

The paper makes substantial theoretical contributions to understanding IQP-QCBM expressivity and trainability. The universality proofs are rigorous and constructive, providing both an asymptotic result and an elegant exact construction requiring only n+1 hidden qubits for an n-qubit generator. The problem is well-motivated through concrete examples demonstrating fundamental limitations of fixed Gaussian kernels, and the generalized MMD framework via spectral representation maintains the crucial property of classical tractability while enabling adaptive kernel learning. The authors demonstrate intellectual honesty by acknowledging fundamental limitations through worst-case analysis, and the overall presentation is clear and well-structured with intuitive explanations accompanying formal results.

**Weaknesses:**

The experimental validation is insufficient to support the practical claims, with evaluation limited to small synthetic datasets specifically constructed to challenge Gaussian kernels, which may artificially inflate the perceived advantage of the adaptive method. The practical utility of the hidden qubit construction remains largely theoretical and unexplored, with no empirical validation of the n+1 hidden qubit universality result and limited analysis of how trainability scales with hidden qubits. The kernel parameterization uses a relatively simple FVSBN architecture without exploration of alternatives or ablation studies, and the initialization strategy appears dataset-specific. Critical experimental details are missing, including computational costs, sensitivity analyses, and proper investigation of the significant overfitting observed in Figure 4. The theoretical analysis, while strong in some areas, lacks finite-sample convergence rates, optimization landscape analysis for the min-max game, and investigation of whether the n+1 hidden qubit bound is tight.

**Questions:**

Lemma 1 and Remark 1: The efficiency claims rely on classical simulation algorithms from prior work (Nest, 2009). More detail on the polynomial dependence on n, m, and ε would strengthen these claims, particularly the hidden constants.

Equation (14): The log-derivative trick for gradient estimation is mentioned but the variance of this estimator isn't analyzed. High variance could make the adversarial training unstable.

Section 5: The worst-case analysis (Lemma 6) constructs pathological distributions with exponentially decaying characteristic functions, but it's unclear how frequently such distributions arise in practice or what structural properties make distributions amenable to MMD-based training.

Overfitting in Figure 4: The appendix shows adaptive kernels achieve much lower training TVD but similar or worse test TVD compared to Gaussian kernels, indicating serious overfitting. This fundamentally undermines the practical utility claims but receives only brief mention.

---

> ### Author Response · Authors · 2025-11-22
>
> We are very grateful to the reviewer for their comments and questions. Please find our responses below.
>
> **W1.**  We agree that the dataset is synthetic. Since our work is the first to introduce methods that go beyond Gaussian kernels for IQP-QCBMs, our priority is to clearly demonstrate potential performance differences. The parity-check dataset was chosen for two reasons:
> -  This is a concrete, practical case where we know precisely why Gaussian fails and the adaptive one can outperform.
> -  It reflects key properties of real-world datasets, e.g., in communication, signal processing, and error correction, where pass/fail labels often arise from parity checks.
>
> We also expect other datasets will, for incidental or opaque reasons, challenge the Gaussian kernel and may only be identifiable by actually training models. Finding such datasets and extending the empirical evaluation to more complex real-world data is an important, complementary direction for future work.
>
> **W2.** The practical advantage of IQP-QCBMs is that the model can be trained on a classical computer with the MMD loss, while deployment/inference requires a quantum computer. We proved that with hidden qubits, we can still train efficiently on classical computers: with $m$ extra hidden qubits and $n$ visible ones, training requires $O(\mathrm{poly}(m+n))$ computation time, compared to $O(\mathrm{poly}(n))$ with visible qubits only. Our universality result with $n+1$ hidden qubits is explicitly a worst-case guarantee for any distribution over $n$ bits. We have now strengthened this: in the revised version, we show that for certain distributions not representable by a visible-only IQP-QCBM, adding just a single hidden qubit already yields an exact representation.
>
> **W3.** The reviewer is right, but this is not necessarily a shortcoming. We introduce a new, general way of defining loss functions for this model, and our key question is whether it can outperform fixed-kernel training, and why. Thus, our focus is not (yet) on designing the “best possible” spectral-measure network, but on showing that learning the spectral measure can overcome fixed-kernel limitations for IQP-QCBMs. Our relatively simple FVSBN parametrisation strengthens this point: even such a simple choice yields advantages. For initialization, we use a single generic Gaussian-based spectral measure, not tuned per dataset.
>
> **W4.** We thank the reviewer for this comment regarding missing critical experimental details. We aim to make the paper as informative as possible and would be very grateful if they could give more details on it.
>
> **Q1.**  We agree that providing the detailed complexity terms improves the statement. We add an explicit bound for it: for $n$ visible qubits, $m$ hidden ones, to guarantee that with at most $\delta$ probability, to estimate the expectation $\langle Z_{\alpha}\rangle$ up to $\varepsilon$ accuracy, the time complexity is lower-bounded by $32\pi^2\mathrm{poly}(n, m)\frac{1}{\varepsilon^2}\log\frac{2}{\delta}$, where $\mathrm{poly}(n, m)$ directly depends on the number of generators that a user choose in IQP-QCBM, e.g., if one choose to use only two-local Pauli-Z operators in the circuit, then $\mathrm{poly}(n, m) \in O((n+m)^2)$. In the revised version, we include the proof in the appendix.
>
> **Q2.**  Indeed, in some cases log derivative trick can incur a high variance of the estimator. However, we numerically checked the empirical variance of the estimated gradient in the training, which is quite small.
>
> **Q3.** We completely agree: the worst-case analysis in Lemma 6 is meant only to show that there exist distributions for which MMD-based training becomes fundamentally hard; it is not intended to model typical data. How often such particular cases occur in practice is an interesting open question beyond the scope of this work.
>
> **Q4.** We would respectfully disagree that Figure 4 showcases overfitting. To explain, a smaller TVD on training/test data sets indicates the generator’s distribution is closer to the target, meaning that the smaller TVD values in Figure 4 the better performance. Figure 4 shows exactly the opposite of what you pointed out. On the training set (the middle row of Figure 4), the adaptive training method is similar to or better than the fixed Gaussian kernel, and on the test set (first row), it is much better than the Gaussian, suggesting that the adaptive training is less prone to overfitting.

---

### Meta-Review · Area_Chair_3stD · 2025-12-26

**Summary:**

This paper studies the Instantaneous Quantum Polynomial Quantum Circuit Born Machine (IQP-QCBM), addressing two key limitations: 1. lack of universality; and  2. instability of MMD-based training with a fixed Gaussian kernel. The authors prove that adding hidden (ancillary) qubits renders IQP-QCBMs universal, and propose a kernel-adaptive MMD training scheme with stronger discriminative power. They provide a theoretical analysis of universality and convergence properties, and validate the adaptive kernel approach on synthetic parity-check datasets, showing improved training behavior compared to fixed kernels.

All four reviewers gave borderline scores, generally acknowledging the theoretical soundness of the universality proof and the novelty of the kernel-adaptive training idea. However, multiple reviewers expressed concerns about the limited empirical evaluation, reliance on synthetic parity-check datasets, and the unclear practical relevance of IQP-QCBMs, especially given hardware noise, scalability, and lack of real-world benchmarks. Several reviewers explicitly noted that, while the theory is interesting, the work risks remaining largely conceptual without stronger evidence of practical utility.

Given the uniformly borderline reviews, I carefully read the paper in detail. My assessment aligns with the reviewers’ concerns: despite theoretical contributions, the paper does not convincingly demonstrate practical applications or near-term relevance of IQP-QCBMs, even with the proposed extensions. Given the high bar for ICLR and the lack of compelling evidence that this framework leads to meaningful practical impact, I recommend rejection.

**Reviewer Concerns:**

The rebuttal addressed several clarification-level concerns raised by the reviewers. In particular, the authors clarified the scope and intent of the universality result, emphasizing that it is meant to establish expressivity in principle rather than practical efficiency at a small scale. They also provided an additional explanation of the kernel-adaptive MMD training scheme.

Two key reviewer concerns remain unresolved after the rebuttal. First, the practical relevance and applications of IQP-QCBMs remain unclear. While the authors emphasize the theoretical nature of the work, the rebuttal does not convincingly demonstrate how the proposed framework translates into meaningful advantages for realistic generative modeling tasks.

Second, the empirical evaluation remains limited. The experiments are restricted to synthetic parity-check datasets, which, while convenient for illustrating theoretical properties, do not provide evidence that the proposed methods are effective in practical generative modeling scenarios. Reviewer concerns about the lack of real-world benchmarks and broader empirical validation therefore remain outstanding.

Beyond the reviewer's concerns, I identified several additional issues upon careful reading. First, the ``classically trained, quantum-deployed'' paradigm is not introduced in this work, and the proposed framework builds heavily on prior work (e.g., arXiv:2503.02934), with much of the theoretical development aimed at addressing known limitations of that model. Related efforts such as arXiv:2509.09033 further reduce the perceived novelty. While the theoretical results are technically sound, the overall contribution does not appear sufficiently distinctive or impactful.

Second, demonstrating sampling advantages on synthetic data has long been a standard paradigm in quantum computing (e.g., random circuit sampling and boson sampling). While such results can establish provable separations, relying solely on synthetic advantages is generally insufficient to attract broad interest, particularly from the classical AI community. This limitation is reflected in several reviewers’ concerns.

Finally, although the authors propose an adaptive kernel to improve the practical performance of IQP-QCBMs, the experimental validation is too limited to assess its effectiveness on realistic tasks. Comparisons against state-of-the-art classical generative models on real-world datasets are necessary to substantiate the practical value of this theoretically motivated approach.

**Reviewer Scores:**

Given the rebuttal and subsequent discussion, it is unlikely that reviewers would have substantially changed their scores. While some clarification-level issues were addressed, the core concerns regarding practical relevance, limited empirical validation, and near-term feasibility of IQP-QCBMs remain unresolved. As a result, reviewers who expressed reservations would likely have maintained their borderline assessments, rather than increasing their scores, after participating fully in the discussion.

---

### Decision · Program_Chairs · 2026-01-26

Reject